# Learning Generalizable Models for Vehicle Routing Problems via Knowledge Distillation

**Jieyi Bi**[1,†] , **Yining Ma**[2,†], **Jiahai Wang**[1,*], **Zhiguang Cao**[3,*] , **Jinbiao Chen**[1],
**Yuan Sun**[4], and **Yeow Meng Chee**[2]

[1]School of Computer Science and Engineering, Sun Yat-sen University
[2]National University of Singapore
[3]Singapore Institute of Manufacturing Technology, A*STAR
[4]University of Melbourne
bijy6@mail2.sysu.edu.cn, yiningma@u.nus.edu
wangjiah@mail.sysu.edu.cn, zhiguangcao@outlook.com
chenjb69@mail2.sysu.edu.cn, yuan.sun@unimelb.edu.au,
ymchee@nus.edu.sg

## Abstract

Recent neural methods for vehicle routing problems always train and test the deep models on the same instance distribution (i.e., uniform). To tackle the consequent cross-distribution generalization concerns, we bring the *knowledge distillation* to this field and propose an *Adaptive Multi-Distribution Knowledge Distillation* (AMDKD) scheme for learning more generalizable deep models. Particularly, our AMDKD leverages various knowledge from multiple teachers trained on exemplar distributions to yield a light-weight yet generalist student model. Meanwhile, we equip AMDKD with an adaptive strategy that allows the student to concentrate on difficult distributions, so as to absorb hard-to-master knowledge more effectively. Extensive experimental results show that, compared with the baseline neural methods, our AMDKD is able to achieve competitive results on both unseen in-distribution and out-of-distribution instances, which are either randomly synthesized or adopted from benchmark datasets (i.e., TSPLIB and CVRPLIB). Notably, our AMDKD is generic, and consumes less computational resources for inference.

## 1 Introduction

The *Vehicle Routing Problem* (VRP) is a class of NP-hard combinatorial optimization problems with a wide variety of practical applications, such as freight delivery [1], last-mile logistics [2] and ride-hailing [3]. For decades, the problem has been studied intensively in computer science and operations research, with numerous exact and (approximate) heuristic algorithms proposed [4–7]. Although the heuristic algorithms are usually preferred in practice given their relatively higher computational efficiency, they heavily rely on hand-crafted rules and domain knowledge, which may still leave room for improvement. As a promising alternative, deep (reinforcement) learning could be used to automatically learn a heuristic (or policy) for VRPs in an *end-to-end* fashion, which has aroused widespread attention in recent years [8–16]. Compared to the traditional ones, the learned heuristics based on deep models could further reduce computational costs while ensuring desirable solutions.

However, existing deep models suffer from inferior generalization with respect to distributions of node coordinates. To be concrete, they often train and test neural networks on instances of the same

---

[†] Equally contributed.
[*] Jiahai Wang and Zhiguang Cao are the corresponding authors.

distribution, mostly the uniform distribution, where deep models are able to achieve competitive results more efficiently than the traditional heuristics (e.g., [12]). Nevertheless, when the learned policy is applied to infer the *out-of-distribution* (OoD) instances, the solution quality is usually low. This cross-distribution generalization issue inevitably hinders the applications of deep models, especially because the real-world VRP instances may follow various and even unknown distributions.

A number of preliminary attempts have been made to tackle this generalization issue for VRPs, which mainly leverage (automatic) data augmentation [17–20] and distributionally robust optimization [21], respectively. However, they are not optimal in our view, as the former always starts with a specified single distribution which may limit the resulted performance, while the latter needs to manually define major and minor instances. Different from them, in this paper, we aim to enhance the cross-distribution generalization by transferring various policies learned from respective distributions into one, where we exploit *knowledge distillation* to learn more generalizable models for VRPs.

Specifically, we propose a generic *Adaptive Multi-Distribution Knowledge Distillation* (AMDKD) scheme for training a light-weight model with favorable cross-distribution generalization performance. To impart broad yet specialized knowledge, we exploit multiple teacher models that are (pre-)trained on respective *exemplar* distributions. Our AMDKD then leverages those teachers to train a shared student model in turns, inspired by the learning paradigm of human-beings. Meanwhile, we also equip the AMDKD with an adaptive strategy to track the real-time learning performance of the student model on every exemplar distribution, which allows the student to concentrate on absorbing those hard-to-master knowledge, so as to strengthen the effectiveness of the learning.

Accordingly, our contributions are summarized as follows: (1) We bring the *knowledge distillation* to the field of *neural combinatorial optimization* and aim at improving the cross-distribution generalization for solving VRPs, which offers a promising perspective to transfer knowledge/policy learned from multiple models into one; (2) We propose the *Adaptive Multi-Distribution Knowledge Distillation* (AMDKD) scheme, where we distill diverse knowledge from multiple teachers trained on exemplar distributions to yield a light-weight yet generalist student model, and also present an adaptive strategy for the student to better assimilate hard-to-master knowledge from difficult distributions; (3) We apply our generic AMDKD to two representative deep models, i.e., AM [10] and POMO [12], respectively. Results show that, while consuming less computational resources for inference, our AMDKD performs favorably against the backbone models as well as other existing generalization methods for deep models, especially on the benchmark dataset CVRPLIB [22]. By further coupling with the efficient active search (EAS) [23], our AMDKD achieves the new state-of-the-art performance. We also conduct a series of analysis to verify our designs.

## 2   Related work

Recently, neural methods based on deep models for VRPs have aroused widespread interest. These methods are categorized as neural *construction* models and neural *improvement* models in general.

**Neural construction models.** They exploit deep (reinforcement) learning to autoregressively construct the solution in an end-to-end fashion. Vinyals et al. [8] introduced the first RNN-based Pointer Network (Ptr-Net) to solve TSP based on supervised learning. Bello et al. [24] then exploited reinforcement learning to train the Ptr-Net for TSP. In [9], the Ptr-Net was then extended to solve CVRP. Different from the above RNN-based methods, Kool et al. [10] proposed the well-known Attention Model (AM) based on the Transformer architecture [25]. Subsequently, many studies extended AM for routing problems, such as [12, 13, 26–30]. As a representative, Kwon et al. [12] proposed the POMO (Policy Optimization with Multiple Optima) and achieved significantly better performance. In a recent work, POMO was adapted to an efficient active search (EAS) framework to further boost the performance during inference [23]. Besides, graph neural networks (GNNs) are also utilized to effectively learn and identify the graph-structured features of the problem instance [1, 11, 31–33].

**Neural improvement models.** They iteratively improve the initial solution by exploiting deep (reinforcement) learning to assist or control the local search. Chen and Tian [34] proposed the NeuRewriter that learned a policy to partially rewrite the current solution. Hottung and Tierney [35, 36] leveraged deep networks to learn to perform the large neighborhood search. Wu et al. [37] and Costa et al. [38] proposed to control the 2-opt operation [6]. Ma et al. [14] further upgraded the deep model of Wu et al. [37] to dual-aspect collaborative Transformer (DACT) with much superior performance. The DACT method was further enhanced in [39] in order to learn a ruin-and-repair

operation for pickup and delivery problems. Generally, improvement methods consume much longer inference time than construction ones to achieve higher solution quality.

**Cross-distribution generalization.** The above neural methods for VRPs often train and test the deep models on the same instance distribution. Although the state-of-the-art deep models perform close to or even surpass the strong traditional heuristics, they are incapable to generalize the learned policy to other distributions, which seriously impairs their practical applications. Therefore, the cross-distribution issue has gradually gained more attention [17–21]. Among the early attempts in this line of research, Xin et al. [18] proposed a generative adversarial network (GAN) based framework to generate hard-to-solve instances for training the model. Wang et al. [20] leveraged the game theory and proposed the Policy Space Response Oracle (PSRO) framework to simultaneously learn a trainable solver and an instance generator. Zhang et al. [17] designed a hardness-adaptive TSP instance generator and adopted curriculum learning to train the model. These methods primarily emphasized on data augmentation or are limited to TSP. Different from them, Jiang et al. [21] adopted CNN to acquire distribution-aware features and exploited group DRO (Distributionally Robust Optimization) to enhance model robustness against distributions. However, this method needs to manually define major and minor instances. Note that we acknowledge that the generalization to different problem scales (or sizes) is also important, which we leave as the future research direction.

## 3  Preliminaries and notations

We first present the formulation of the studied VRPs and classic distributions. Then we introduce how deep models are used to solve them, followed by the basic rationale of knowledge distillation.

### 3.1  VRPs and their distributions

We define VRPs over a complete graph $\mathcal{G} = \{\mathcal{V}, \mathcal{E}\}$, where $v_i \in \mathcal{V}$ represents the (customer) node, $e(v_i, v_j) \in \mathcal{E}$ represents the edge between two nodes, and $C[e(v_i, v_j)]$ represents the cost (we use *length* in this paper) of the edge. By referring tour $\tau$ (a.k.a. solution) to a permutation of nodes in $\mathcal{V}$, the objective is usually to find the optimal tour $\tau^*$ with the least total cost(length) over a finite search space $S$ containing all possible tours, which could be formulated as Eq. (1) in general,

$$\tau^* = \underset{\tau' \in \mathcal{S}}{\arg\min}\, L(\tau'|\mathcal{G}) = \underset{\tau' \in \mathcal{S}}{\arg\min} \sum_{e(v_i, v_j) \in \tau'} C\left[e(v_i, v_j)\right]. \tag{1}$$

For different VRP variants, such objective may be subject to different problem-specific constraints [40, 41], where multiple sub-tours may exist in a valid tour $\tau$. Following the recent literature [10, 12, 23], we focus on two representative VRPs, i.e., TSP and CVRP, respectively. A feasible tour for TSP considers a vehicle visiting each node in $\mathcal{V}$ once and only once. As an extension from a vehicle to a fleet, CVRP considers an extra depot node $v_0$ and a capacity limit $Q$ for any given vehicle, where each customer node $v_i(i = 1, ..., n)$ is associated with a demand request $\delta_i$. A feasible tour for CVRP consists of multiple sub-tours, each of which represents a vehicle in the fleet departing from the depot, serving a subset of nodes, and finally returning to the depot. The total demand of each sub-tour must not exceed the capacity $Q$, and all nodes except for the depot must be visited once and only once.

In Figure 1, we visualize a number of VRP instances following various distributions from the literature [10, 21, 43], including the TSPLIB [42] and CVRPLIB [22] benchmark datasets. As can be observed, the node coordinates of a VRP instance may follow complicated and even unknown distribution, which considerably intensifies the hardness for solving. While the recent neural methods report superior performance to the traditional heuristics on some fixed distributions, it is unfortunate that they are more sensitive to distribution shift. It is thus of great importance and interest to develop powerful deep models that can simultaneously handle as many diverse distributions as possible.

### 3.2  Tour construction by deep models

We focus on neural construction methods for VRPs, which usually exploit deep neural networks to sequentially construct the tour. In light of this, the solving procedure is mostly modelled as a Markov Decision Process (MDP), where the Transformer styled [10] architectures following the encoder-decoder structure are often adopted as the policy network. Typically, the encoders project the nodes of the instance into node embeddings for feature extraction. Afterwards, the decoder builds the

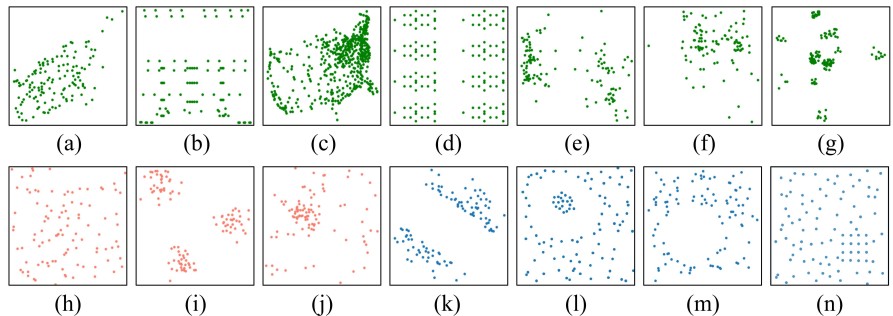

Figure 1: VRP instances following various distributions from the literature: (a) gr137, (b) lin105, (c) att532, (d) pr136, (e) X-n125-k30, (f) bier127, (g) Tai150d, (h) Uniform, (i) Cluster, (j) Mixed, (k) Expansion, (l) Implosion, (m) Explosion, (n) Grid, where instances (a)-(g) are from TSPLIB [42] and CVRPLIB [22]. In this paper, we consider instances following distributions (h)-(j) for training and other unseen distributions (k)-(n), as well as unseen benchmark datasets for testing.

tour $\tau$ based on learned node embeddings and the partial tour $a_{1:t-1}$ constructed previously. At time step $t$ of the MDP, the decoder picks an unvisited node as the action $a_t$ where invalid ones are masked for feasibility. The procedure is repeated until the whole tour is completed, which is factorized as,

$$p_\theta(\tau|\mathcal{G}) = \prod_{t=1}^{\ell} p_\theta(a_t|a_{1:t-1}, \mathcal{G}), \tag{2}$$

where $p_\theta$ is the policy parameterized by $\theta$, and $\ell$ denotes the final time step in MDP. For TSP, $\ell = n$; for CVRP, $\ell \geq n$ since the depot can be visited more than once[1]. The total reward is defined as the negative of the tour length, i.e., $-L(\tau|\mathcal{G})$. This is essentially consistent with the objective in Eq. (1).

### 3.3 Knowledge distillation

Knowledge distillation [44] is a kind of teacher-student training paradigm that aims at transferring knowledge from a (group of) complex teacher model(s) $\theta^{\mathrm{T}}$, to a succinct student model $\theta^{\mathrm{S}}$. The recent research findings reveal that knowledge distillation is not only able to effectively learn a lighter student network from larger teacher network(s) [45, 46], but also has potential to improve the generalization [47, 48] even over its teacher(s) [49–51]. Typically, the teacher models are pre-trained, and the student model compares the output of teacher model(s) with its own and considers it as a supervisory signal. Formally, the student network is trained with the goal of minimizing a weighted combination of the distillation loss $\mathcal{L}_{\mathrm{KD}}$ and the original task loss $\mathcal{L}_{\mathrm{Task}}$ as follows,

$$\mathcal{L} = \alpha \mathcal{L}_{\mathrm{Task}} + (1 - \alpha)\mathcal{L}_{\mathrm{KD}}, \tag{3}$$

where $\alpha \in [0, 1]$. The $\mathcal{L}_{\mathrm{KD}}$ with respect to one or more teachers ($N_T \geq 1$) is generally formulated as,

$$\mathcal{L}_{\mathrm{KD}} = \frac{1}{N_T} \sum_{x \in \mathcal{X}} \sum_{i=1}^{N_T} \phi \left[ p_{\theta_i^{\mathrm{T}}}(x), p_{\theta^{\mathrm{S}}}(x) \right], \tag{4}$$

where $x$ is the training data from $\mathcal{X}$ and $\phi(\cdot)$ measures the statistical distance between the teacher and the student, such as the Kullback-Leibler divergence $\mathrm{KL}(p_{\theta^{\mathrm{T}}} \| p_{\theta^{\mathrm{S}}}) = \sum_i p_{\theta^{\mathrm{T}}}(\log p_{\theta^{\mathrm{T}}} - \log p_{\theta^{\mathrm{S}}})$.

## 4 Methodology

Consider what happens in a classroom when students study multiple modules. Typically, they (students) learn only one module (exemplar distribution) at a time from a professional teacher, and then go on to the next one until all modules are mastered. When students do poorly on a quiz after class (validation dataset), they will be required to study that module more frequently in order to

---

[1]The sub-tours for CVRP are constructed sequentially. At each step, the agent selects an unvisited node whose demand is smaller than the remaining capacity or returns to the depot for full replenishment.

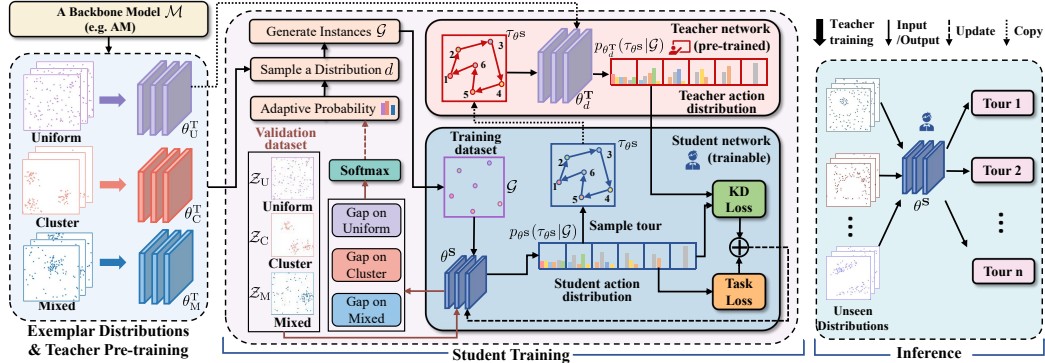

Figure 2: Framework of our AMDKD scheme. From left to right: teacher pre-training, student training and inference. The Uniform is selected as the current exemplar distribution for an example.

become generalists. Motivated by this, we propose the *Adaptive Multi-Distribution Knowledge Distillation* (AMDKD) scheme that could efficiently learn more generalizable models for VRPs.

Previously, deep models were trained on a single distribution, e.g., the Uniform in [12–14]. More recently, attempts have been made to expand the training data by augmenting atypical (minor) [21] or hard instances [17, 18]. Differently, AMDKD allows expert knowledge for tackling distinct distributions to be transferred into a single yet generalist model via distillation. With a light network and high inference speed, the resulted model is supposed to perform favorably against existing methods. Moreover, AMDKD is generic, and could be used to boost various deep models for VRPs.

## 4.1 Overall structure and teacher (pre)-training

The overview framework of our AMDKD is illustrated in Figure 2. Given an existing deep model (e.g., AM [10]), AMDKD first performs teacher training (or directly use pre-trained ones) to obtain a teacher model for each distribution listed in a set of *exemplar* ones. Then in each knowledge transfer epoch, AMDKD picks a specific distribution $d$ and its teacher $\theta_d^{\mathrm{T}}$ to train a student network. To facilitate effective learning, the likelihood of picking each distribution is adaptively updated according to the current performance of the student on a validation dataset. Finally, AMDKD performs on-policy distillation, allowing the student network $\theta^{\mathrm{S}}$ to sample RL trajectories (i.e., tours $\{\tau_{\theta^{\mathrm{S}}}\}$) for training, and calculates the two losses (i.e., KD loss and Task loss) for an update.

We consider Uniform, Cluster, and Mixed (mixture of uniform and cluster) as exemplar distributions to train teacher networks in this paper. Consequently, a set of well-trained teachers with parameters $\theta^{\mathrm{T}} = \{\theta_{\mathrm{U}}^{\mathrm{T}}, \theta_{\mathrm{C}}^{\mathrm{T}}, \theta_{\mathrm{M}}^{\mathrm{T}}\}$ are attained for each distribution. Despite only limited distributions are exploited throughout training, the learned student network is expected to absorb generic and robust knowledge from teachers that may effectively generalize to unseen distributions such as Expansion, Implosion, Explosion, and Grid (visualization in Figure 1). Our motivation here is that each teacher only needs to master a specific distribution, and by working together via our distilling scheme, they can educate a generalist student. Note that the exemplar distributions used here could also be substituted with others (see Section 5.3). Furthermore, since our designed distillation scheme is model-agnostic, the teacher network can follow most of existing architectures. In this paper, we assess our scheme by applying it to two representative construction methods, i.e., AM [10] and POMO [12], respectively.

## 4.2 Adaptive multi-distribution student training

Given the well-trained teachers on exemplar distributions, our AMDKD student selects one teacher and one distribution in each knowledge transfer epoch and gradually learns to make appropriate decisions. Note that there is another line of works that argue for the simultaneous use of multiple teachers in the distillation process [52, 53]. In their strategy, all teachers are engaged to provide a weighted loss to train the student as aforementioned in Section 3.3. However, such design may require teachers to be trained on a homogeneous task primarily with supervised learning, which does not suit our heterogeneous distributions with reinforcement learning (see Appendix C for results).

---
**Algorithm 1** Adaptive Multi-Distribution Knowledge Distillation (AMDKD)
---
**Input:** A backbone model $\mathcal{M}$ (e.g., AM), exemplar distributions $\mathcal{D}$ (e.g., $\mathcal{D} = \{U, C, M\}$).
1: Randomly initialize teacher networks $\theta_d^{\mathrm{T}}$ ($\forall d \in \mathcal{D}$) and student network $\theta^{\mathrm{S}}$ according to $\mathcal{M}$;
2: Perform teacher training or leverage pre-train ones (if any) to attain well-learned $\theta_d^{\mathrm{T}}$ ($\forall d \in \mathcal{D}$);
3: **for** epoch $= 1, 2, ..., E$ **do**
4:     Pick distribution $d$ and its teacher $\theta_d^{\mathrm{T}}$ with an adaptive probability according to Eq. (5);
5:     **for** step $= 1, 2, ..., T$ **do**
6:         Let student $\theta^{\mathrm{S}}$ sample tours $\tau_{\theta^{\mathrm{S}}}^i$ for each $\{\mathcal{G}_i\}_{i=1}^B$ according to its own policy $p_{\theta^{\mathrm{S}}}$;
7:         Get $\nabla\mathcal{L}_{\mathrm{Task}}$ by estimating Eq. (6) as per the original design of $\mathcal{M}$;
8:         Get $\nabla\mathcal{L}_{\mathrm{KD}}$ by computing the gradients of Eq. (7);
9:         $\theta^{\mathrm{S}} \leftarrow \theta^{\mathrm{S}} + \eta\nabla\mathcal{L}$ where $\nabla\mathcal{L} \leftarrow \alpha\nabla\mathcal{L}_{\mathrm{Task}} + (1 - \alpha)\nabla\mathcal{L}_{\mathrm{KD}}$.
10:    **end for**
11: **end for**
---

**Student network.** We stipulate that the student shares a similar architecture with its teachers, but can reduce its network parameters as needed to speed up the inference. Nevertheless, we find in Section 5.3 that a larger student model usually leads to a better performance. Therefore, there is a trade-off between solution quality and computational cost. In this paper, we consider reducing the dimension of the node embeddings from $128$ (teacher) to $64$ (student), resulting in a reduction of $61.8\%$ and $59.2\%$ in the model parameters for the adaption with AM and POMO, respectively. We also considered lowering the number of encoder layers, but the results were below the expectation.

**The adaptive multi-distribution distilling strategy.** In each epoch, AMDKD selects one distribution and its corresponding teacher model. At the beginning, it selects each distribution with an equal probability. After epoch $E'$ (a hyper-parameter), the probability would be adaptively adjusted according to the performance of the student on the given validation datasets $\mathcal{Z}_{\mathrm{U}}, \mathcal{Z}_{\mathrm{C}}, \mathcal{Z}_{\mathrm{M}}$ (each with 1,000 instances) for each exemplar distribution. The likelihood $p^{\mathrm{adaptive}}$ of selecting distribution $d \in \{U, C, M\}$ is proportional to the exponent value of the gaps to the LKH solver [4] as follows,

$$p^{\mathrm{adaptive}}(d) = \begin{cases} \mathrm{Softmax}\left(\mathrm{AvgGap}\left[\{\tau_{\theta^{\mathrm{S}}}\}|_{\mathcal{Z}_d}, \{\tau_{\mathrm{solver}}\}|_{\mathcal{Z}_d}\right]\right), & \text{if } E \geq E' \\ \frac{1}{|\mathcal{D}|}, & \text{otherwise} \end{cases} \quad (5)$$

where $|\mathcal{D}|$ is the number of exemplar distributions; $\{\tau_{\theta^{\mathrm{S}}}\}|_{\mathcal{Z}_d}$ and $\{\tau_{\mathrm{solver}}\}|_{\mathcal{Z}_d}$ refer to the tour set generated by the student $\theta^{\mathrm{S}}$ and the LKH solver on dataset $\mathcal{Z}_d$, respectively. The validation sets are fixed, hence the LKH solver only needs to run once to attain $\{\tau_{\mathrm{solver}}\}|_{\mathcal{Z}_d}$ for $d \in \{U, C, M\}$. Meanwhile, we note that $p^{\mathrm{adaptive}}$ may converge during distillation (see Appendix C), which means that it is possible to stop such evaluation early and reuse the stabilized one for an even faster training.

**Loss function.** Following Eq. (3), we leverage the task loss ($\mathcal{L}_{\mathrm{Task}}$) and the KD loss ($\mathcal{L}_{\mathrm{KD}}$) to jointly train the student with $\alpha = 0.5$. Pertaining to $\mathcal{L}_{\mathrm{Task}}$, we define the task loss as follows,

$$\mathcal{L}_{\mathrm{Task}} = -J(\theta^{\mathrm{S}}|d) = -\mathbb{E}_{\mathcal{G}\sim d, \tau\sim p_{\theta^{\mathrm{S}}}(\tau|\mathcal{G})}[L(\tau|\mathcal{G})], \quad (6)$$

where the training instances $\mathcal{G}$ are sampled following the selected distribution $d$, and the tours $\tau$ are constructed via the student network $\theta^{\mathrm{S}}$ according to Eq. (2). In AM and POMO, the REINFORCE algorithm [54] is used to estimate the gradients for the above loss function and we follow exactly the same way as per their original designs. Pertaining to $\mathcal{L}_{\mathrm{KD}}$, it is defined to encourage the student to imitate how a teacher network sequentially selects the nodes. Specifically, given the instance $\mathcal{G}$ sampled from distribution $d$ and a tour $\tau_{\theta^{\mathrm{S}}}$ constructed by the student $\theta^{\mathrm{S}}$ following its own policy, our AMDKD leverages $p_{\theta_d^{\mathrm{T}}}(\tau_{\theta^{\mathrm{S}}}|\mathcal{G})$ suggested by the teacher network $\theta_d^{\mathrm{T}}$, to compute $\mathcal{L}_{\mathrm{KD}}$ that measures the similarity of the probability distributions between teacher and student using the KL divergence,

$$\mathcal{L}_{\mathrm{KD}} = \frac{1}{B}\sum_{i=1}^B \sum_{a_j \in \tau_{\theta^{\mathrm{S}}}} p_{\theta_d^{\mathrm{T}}}(a_j|\mathcal{G}_i)\left(\log p_{\theta_d^{\mathrm{T}}}(a_j|\mathcal{G}_i) - \log p_{\theta^{\mathrm{S}}}(a_j|\mathcal{G}_i)\right). \quad (7)$$

We summarize our AMDKD in Algorithm 1. Note that our AMDKD follows the on-policy scheme where the tours $\tau_{\theta^{\mathrm{S}}}$ are output by the student that currently performs learning. As an alternative, such tours can also be output by the teacher (i.e., off-policy scheme). However, it performs inferior to our on-policy one (see Section 5.3). Finally, we note that the student learned by our AMDKD could also be coupled with the *efficient active search* (EAS) [23] during inference to further boost the performance. Accordingly, the resulting AMDKD+EAS achieves a new state-of-the-art performance.

Table 1: Distillation effectiveness of AMDKD on three exemplar distributions.

| | Model | Size (M) | n = 20 | | | | n = 50 | | | | n = 100 | | | |
|---|---|---|---|---|---|---|---|---|---|---|---|---|---|---|
| | | | $G_U$ | $G_C$ | $G_M$ | Avg. | $G_U$ | $G_C$ | $G_M$ | Avg. | $G_U$ | $G_C$ | $G_M$ | Avg. |
| TSP | AM(U) | 0.68 | 0.09% | 0.26% | 0.19% | 0.18% | 0.59% | 2.24% | 1.36% | 1.39% | 2.10% | 7.49% | 4.06% | 4.55% |
| | AM(C) | 0.68 | 0.17% | 0.10% | 0.27% | 0.18% | 1.41% | 0.80% | 2.14% | 1.45% | 3.76% | 6.97% | 4.39% | 5.04% |
| | AM(M) | 0.68 | 0.15% | 0.16% | 0.13% | 0.15% | 1.19% | 1.71% | 0.87% | 1.26% | 3.08% | 5.65% | 2.55% | 3.76% |
| | AMDKD-AM | **0.26** | 0.02% | 0.06% | 0.05% | **0.04%** | 0.25% | 1.64% | 0.86% | **0.91%** | 1.21% | 5.63% | 3.55% | **3.46%** |
| | POMO(U) | 1.20 | 0.00% | 0.01% | 0.01% | 0.01% | 0.04% | 0.42% | 0.21% | 0.22% | 0.17% | 1.97% | 0.92% | 1.02% |
| | POMO(C) | 1.20 | 0.00% | 0.00% | 0.01% | 0.00% | 0.09% | 0.07% | 0.21% | 0.12% | 0.41% | 0.29% | 0.83% | 0.51% |
| | POMO(M) | 1.20 | 0.00% | 0.01% | 0.00% | 0.00% | 0.08% | 0.17% | 0.08% | 0.11% | 0.77% | 1.17% | 0.34% | 0.76% |
| | AMDKD-POMO | **0.49** | 0.00% | 0.00% | 0.00% | **0.00%** | 0.05% | 0.05% | 0.09% | **0.06%** | 0.34% | 0.35% | 0.41% | **0.37%** |
| CVRP | AM(U) | 0.68 | 1.98% | 1.99% | 1.98% | 1.98% | 2.53% | 4.33% | 2.99% | 3.28% | 3.10% | 9.87% | 4.57% | 5.85% |
| | AM(C) | 0.68 | 1.62% | 1.43% | 1.74% | 1.60% | 3.08% | 2.75% | 3.35% | 3.06% | 4.27% | 3.89% | 4.93% | 4.36% |
| | AM(M) | 0.68 | 2.09% | 2.19% | 2.05% | 2.11% | 2.74% | 3.17% | 2.31% | 2.74% | 3.95% | 6.26% | 3.41% | 4.54% |
| | AMDKD-AM | **0.26** | 0.53% | 0.59% | 0.64% | **0.59%** | 1.61% | 2.66% | 1.92% | **2.07%** | 2.08% | 5.06% | 3.01% | **3.38%** |
| | POMO(U) | 1.20 | 0.36% | 0.49% | 0.51% | 0.45% | 0.80% | 1.53% | 1.07% | 1.13% | 0.95% | 2.34% | 1.31% | 1.53% |
| | POMO(C) | 1.20 | 0.41% | 0.40% | 0.54% | 0.45% | 1.16% | 0.93% | 1.07% | 1.05% | 0.93% | 1.28% | 1.21% | 1.14% |
| | POMO(M) | 1.20 | 0.36% | 0.51% | 0.40% | 0.42% | 1.22% | 1.34% | 0.85% | 1.14% | 1.89% | 2.07% | 0.96% | 1.64% |
| | AMDKD-POMO | **0.49** | 0.35% | 0.40% | 0.41% | **0.39%** | 0.81% | 0.97% | 0.89% | **0.89%** | 1.06% | 1.36% | 0.99% | **1.13%** |

*Note:* Unless otherwise stated, the gaps are computed w.r.t. the strong traditional solvers Gurobi [5] (for TSP) and LKH [4] (for CVRP).

# 5 Experiments

We conduct experiments on TSP and CVRP with $n = 20$, 50, and 100 nodes similar to [12, 14]. As aforementioned, we adopt Uniform, Cluster and Mixed (mixture of uniform and cluster) as exemplar distributions for training; Expansion, Implosion, Explosion, and Grid as the unseen distributions for testing. For the above 7 distributions, we follow [10, 21, 43] to generate the respective instances (details are presented in Appendix A). All experiments are conducted on a machine with NVIDIA RTX 3090 GPU cards and Intel Xeon Silver 4216 CPU at 2.10GHz. Our implementation in PyTorch are publicly available[2]. Some additional analysis and discussions can be found in Appendix C.

**Training and hyper-parameters.** We apply our AMDKD to two representative deep models, i.e., AM [10] and POMO [12], termed as AMDKD-AM and AMDKD-POMO, respectively. For the teacher (pre)-training phase, we directly follow the RL algorithm, network architecture, and other hyper-parameters as suggested in the original backbone method. For the student distillation phase, we use batch size $B = 512^3$, and task-specific hyper-parameters $T = 250$, $E' = 500$ for AMDKD-AM and $T = 20$, $E' = 1$ for AMDKD-POMO, respectively. By default, the dimension of the node embeddings in our student networks AMDKD-AM and AMDKD-POMO is reduced from 128 (teacher) to 64 (student) to enjoy a faster inference speed. The total numbers of needed training epochs vary with the problem size. Regarding AMDKD-AM, we use 2,000, 5,000, 10,000 for problem sizes 20, 50, 100, respectively (both TSP and CVRP); regarding AMDKD-POMO, we use 5,000 (TSP-20), 10,000 (CVRP-20, TSP-50, TSP-100), and 30,000 (CVRP-50, CVRP-100). The Adam optimizer is used with learning rate 1e-4. Training time also varies with the problem size. Taking CVRP-100 as an example, one epoch takes about 4 minutes for AMDKD-AM and 1.4 minutes for AMDKD-POMO.

**Inference.** For AMDKD-AM, it samples 1,280 solutions following AM [10]; and for AMDKD-POMO, we adopt the greedy rollout with $\times 8$ augments following POMO [12]. All experiments are conducted on test datasets containing 10,000 instances per distribution. Unless otherwise stated, the gaps are computed w.r.t. the strong traditional solvers Gurobi [5] (for TSP) and LKH [4] (for CVRP).

## 5.1 Effectiveness analysis of AMDKD

We first study the effectiveness of our AMDKD when applied to backbone model AM and POMO for TSP and CVRP, respectively. In Table 1, we display the gaps of the learned student and their respective teachers on unseen instances following three exemplar distributions (denoted as $G_U$, $G_C$, and $G_M$) and also report the parameter sizes of each model. We can see that when the teacher model is trained on a specific distribution, it does not generalize well on the other ones (especially for AM), resulting in poor overall performance (if referring to the average gaps). In contrast, our AMDKD could alleviate this issue effectively. While the learned student model by AMDKD reduces the size of

---
[2] https://github.com/jieyibi/AMDKD
[3] For training AMDKD-POMO on TSP100 and CVRP100, we use $B = 128$ due to GPU memory constraint.

Table 2: Generalization on unseen in-distribution (ID) and out-of-distribution (OoD) instances.

| | Model | Size (M) | $n=20$ $G_{ID}$ | $G_{OoD}$ | Time* | $n=50$ $G_{ID}$ | $G_{OoD}$ | Time* | $n=100$ $G_{ID}$ | $G_{OoD}$ | Time* |
|---|---|---|---|---|---|---|---|---|---|---|---|
| **TSP** | Gurobi | - | - | - | 0.01s (7s) | - | - | 0.08s (51s) | - | - | 0.7s (7.6m) |
| | HAC[#] | 0.68 | 0.11% | 0.07% | 0.03s (48s) | 1.05% | 0.57% | 0.09s (4m) | 4.68% | 2.97% | 0.19s (16m) |
| | LCP[#] | 2.03 | 0.11% | 0.03% | 1.2s (43m) | 6.70% | 0.99% | 1.5s (2.8h) | 32.70% | 8.24% | 2.4s (6.4h) |
| | DACT(T=1,280)[#] | 0.27 | 0.11% | 0.08% | 12s (2.1m) | 0.31% | 0.34% | 19s (7.2m) | 2.37% | 3.13% | 28s (23m) |
| | AM[#] (128) | 0.68 | 0.18% | 0.10% | 0.03s (48s) | 1.48% | 0.74% | 0.09s (4m) | 4.15% | 2.84% | 0.19s (16m) |
| | AM[#] (64) | 0.26 | 0.21% | 0.11% | 0.03s (35s) | 1.74% | 0.83% | 0.08s (2.8m) | 5.93% | 3.85% | 0.17s (11m) |
| | AMDKD-AM (64) | 0.26 | **0.04%** | **0.02%** | 0.03s (35s) | **0.91%** | **0.37%** | 0.08s (2.8m) | **3.46%** | **1.87%** | 0.17s (11m) |
| | POMO[#] (128) | 1.20 | 0.00% | 0.00% | 0.03s (5s) | 0.07% | 0.05% | 0.09s (16s) | **0.30%** | **0.28%** | 0.13s (1.1m) |
| | POMO[#] (64) | 0.49 | 0.02% | 0.01% | 0.02s (4s) | 0.18% | 0.16% | 0.04s (11s) | 0.69% | 0.59% | 0.12s (50s) |
| | AMDKD-POMO (64) | 0.49 | **0.00%** | **0.00%** | 0.02s (4s) | **0.06%** | **0.05%** | 0.04s (11s) | 0.37% | 0.41% | 0.12s (50s) |
| | AMDKD+EAS[†] | 0.49 | **0.00%** | **0.00%** | 5s (4.5m) | **0.01%** | **0.01%** | 12s (28m) | **0.11%** | **0.10%** | 28s (2.3h) |
| **CVRP** | LKH3 | - | - | - | 7.7s (1.3h) | - | - | 31s (5.3h) | - | - | 56s (9.6h) |
| | DACT(T=1,280)[#] | 0.27 | 0.09% | 0.05% | 28s (4.3m) | 1.59% | 1.59% | 55s (14m) | 5.56% | 5.55% | 1.5m (34m) |
| | AM[#] (128) | 0.68 | 2.00% | 1.98% | 0.05s (1.2m) | 3.39% | 2.66% | 0.12s (5m) | 5.42% | 3.75% | 0.26s (18m) |
| | AM[#] (64) | 0.26 | 2.02% | 2.02% | 0.05s (49s) | 3.62% | 2.65% | 0.11s (3.6m) | 6.83% | 4.56% | 0.23s (13m) |
| | AMDKD-AM (64) | 0.26 | **0.59%** | **0.55%** | 0.05s (49s) | **2.07%** | **1.69%** | 0.11s (3.6m) | **3.38%** | **2.44%** | 0.23s (13m) |
| | POMO[#] (128) | 1.20 | 0.42% | 0.39% | 0.05s (7.8s) | 0.92% | 0.94% | 0.10s (18s) | 1.14% | 1.21% | 0.19s (1.3m) |
| | POMO[#] (64) | 0.49 | 0.55% | 0.51% | 0.05s (6.1s) | 1.19% | 1.21% | 0.08s (15s) | 1.43% | 1.50% | 0.18s (1.1m) |
| | AMDKD-POMO (64) | 0.49 | **0.39%** | **0.36%** | 0.05s (6.1s) | **0.89%** | **0.90%** | 0.08s (15s) | **1.13%** | 1.21% | 0.18s (1.1m) |
| | AMDKD+EAS[†] | 0.49 | **-0.04%** | **-0.06%** | 9s (7.8m) | **0.07%** | **0.04%** | 20s (37m) | **-0.03%** | **-0.04%** | 40s (3.3h) |

* We report the average time to solve one instance, and the total time to solve 10,000 instances in (·) with batch parallelism allowed (one GPU).
[#] The corresponding model is trained on a mixed training dataset that contains instances from all the three exemplar distributions.
[†] For EAS, we adopt its EAS-lay version (T=100) for demonstration purpose.

the teacher model from 0.68 to 0.26 M (a 61.8% reduction) for AM and from 1.20 to 0.49 M (a 59.2% reduction) for POMO, it still exhibits improved overall performance for both TSP and CVRP on all the three sizes. Pertaining to AMDKD-AM, our AMDKD student not only significantly outperforms its three teachers for both TSP and CVRP, but also exhibits even lower gaps on the distribution where the teacher was previously trained in most cases. Pertaining to AMDKD-POMO, although POMO itself presents a better generalization, our AMDKD can still boost its overall performance with a much lighter student network for both TSP and CVRP. The above results validate that our AMDKD does not overfit to a single distribution, and successfully learns a lightweight yet generalist student model under the guidance of multiple teachers with expertise in different exemplar distributions.

## 5.2 Generalization analysis of AMDKD

We now assess the cross-distribution generalization of our AMDKD. In addition to the two backbone models, we also compare with the following methods, i.e., 1) other deep models including- 1.a) DACT [14], a neural *improvement* model that learns to perform local search; 1.b) LCP [13] (TSP only), a hybrid two-stage method that combines *improvement* and *construction* strategy; and 2) other methods specialized for generalization including- 2.a) HAC [17] (TSP only), a fine-tuning framework for AM by leveraging instances with different hardness; 2.b) DROP [21], a *distributionally robust optimization* based method to enhance POMO; 2.c) GANCO [18], a framework that enhances AM by learning a *generative adversarial network* to produce hard-to-solve training instances; and 2.d) PSRO (a.k.a. LIH) [20], an *improvement* method with a *game theory* based policy space response oracle framework. More implementation details of them are provided in Appendix B. Note that for DROP, GANCO, and PSRO, we only compare the results on several benchmark instances reported in their original papers (see Table 3), since their codes (for re-training) are not publicly available.

**Distribution mixture augmentation.** Given that baselines AM, POMO, DACT, and LCP were originally trained on the uniform distribution only (resulting in poor generalization), we re-train the above models (with #) on a mixed dataset containing instances from the three exemplar distributions, to ensure a fair comparison. We also apply this mixed dataset as the initial training instances for HAC (with #). Though such distribution mixture augmentation improves the generalization of construction methods POMO and AM on larger instances, we notice that it does not always happen so, especially for the methods that have an improvement component (e.g., DACT and LCP). We present detailed results and discussions regarding this finding in Appendix B. Meanwhile, for AM[#] and POMO[#], we report their performance with network dimensions of both 128 and 64 (same size as our AMDKD student) for a more comprehensive comparison. We assess the following metrics, 1) parameter size; 2)

Table 3: Generalization performance on selected instances ($100 \leq n \leq 200$) from benchmark datasets.

| | PSRO | AM (128) | GANCO | HAC | AM$^{\#}$(128) | AMDKD-AM (64) | POMO (128) | DROP | POMO$^{\#}$(128) | AMDKD-POMO (64) | AMDKD+EAS |
|---|---|---|---|---|---|---|---|---|---|---|---|
| TSPLIB | 4.47% | 42.63% | 4.87% | 6.06% | 17.60% | **3.53%** | 29.73% | 10.79% | **0.87%** | 1.08% | **0.74%** |
| CVRPLIB | - | 29.36% | - | - | 13.88% | **7.43%** | 14.19% | 8.67% | 6.80% | **4.38%** | **1.26%** |

average gap on unseen in-distribution instances, i.e., $G_{ID}$; 3) average gap on unseen out-of-distribution instances, i.e., $G_{OoD}$; 4) average time for solving an individual instance and total time for solving 10,000 instances with batch parallelism allowed (one GPU)[4].

As can be observed in Table 2, our AMDKD is able to achieve competitive results on both unseen in-distribution and out-of-distribution instances, and performs favourably against all baselines. Pertaining to TSP, our AMDKD-AM (64) significantly outperforms the baseline AM$^{\#}$ (128) and AM$^{\#}$ (64) on all sizes in terms of both $G_{ID}$ and $G_{OoD}$ with fewer parameters and higher inference speed. Meanwhile, it also consistently beats the recent baseline HAC$^{\#}$ (designed to boost the generalization of AM) on all sizes in terms of both two gaps and the time efficiency. For the backbone model POMO, which suffers from less generalization concern, the baseline POMO$^{\#}$ (128) can already attain near-optimal solutions to TSP instances with a desirable generalization performance. However, our AMDKD (64) is able to achieve competitive generalization results with nearly half of the model size and higher inference speed. When we reduce the network dimension of POMO$^{\#}$ to 64 (as our AMDKD), POMO$^{\#}$ (64) performs much inferior to our AMDKD-POMO (64). Pertaining to CVRP, similar patterns can be observed, where our AMDKD-AM (64) and AMDKD-POMO (64) consistently deliver favourable generalization performance against all the compared baselines. Besides, both our AMDKD-AM (64) and AMDKD-POMO (64) exhibit significantly faster inference than improvement methods LCP and DACT while achieving similar or even lower gaps.

Furthermore, we show that by coupling our AMDKD-POMO (64) model with the recent *efficient active search* (EAS) [23], the resulting AMDKD+EAS ($T$=100) attains considerably superior performance that even surpasses LKH3 on CVRP-100 with much shorter time. Given the advances of active search for a particular instance to be inferred, AMDKD+EAS allows the model learned by our AMDKD with strong generalization to continuously improve its performance during inference (longer $T$ will yield even better performance), which leads to a new state-of-the-art hybrid solver.

We now evaluate the generalization of our AMDKD on the benchmark, i.e., TSPLIB and CVRPLIB, which contain various instances of unknown distributions and larger size. As shown in Table 3, our AMDKD-AM (64) significantly boosts the generalization of AM (128) and also outperforms the baseline AM$^{\#}$ (128) as well as other generalization methods that take AM (128) as the backbone model (GANCO and HAC). Similarly, our AMDKD-POMO (64) also significantly beats the other baseline methods, except for POMO$^{\#}$ (128) on TSPLIB. Nevertheless, regarding the harder CVRP instances from CVRPLIB, our AMDKD-POMO (64) could yield significantly better performance. Finally, we note that the AMDKD+EAS achieves the smallest gaps for both TSP and CVRP among all baselines. We refer to Appendix D for full results and discussions.

## 5.3 Further analysis of AMDKD

**Effects of different components.** We conduct ablation studies on CVRP-50 and TSP-50 to verify the designs of our AMDKD, where we consider removing the proposed adaptive strategy, the KD loss $\mathcal{L}_{KD}$, and the task loss $\mathcal{L}_{Task}$, respectively. As displayed in Table 4, the use of adaptive strategy further enhances the learning effectiveness, and the two losses play a prominent role in the phase of gradient update. We also verify that when compared with our on-policy scheme, the off-policy variant (as mentioned in Section 4.2) saliently impairs the distillation performance.

**Effects of student network architectures.** In Figure 3, we exhibit the boxplots of the averaged gaps when using different student network architectures (different dimensions of the entire network and different number of layers in its encoder) of AMDKD-POMO for solving CVRP-50. As revealed, the larger the model, the better its performance. Hence, we acknowledge that our AMDKD can be further boosted when the student (64) shares the same architecture with its teachers (128). Specifically, for CVRP-50, our AMDKD-POMO (128) exhibits an average optimality gap of 0.81%, which is

---

[4]Even so, we note that the time of traditional solvers and deep models might be still difficult to be fairly compared due to different implementations (C vs Python) and computing devices (CPU vs GPU).

Table 4: Ablation studies of AMDKD designs.

| | | Components | | | CVRP-50 | TSP-50 |
|---|---|---|---|---|---|---|
| Model | $p^{\text{adaptive}}$ | $\mathcal{L}_{\text{KD}}$ | $\mathcal{L}_{\text{Task}}$ | $\tau_\theta$s | Avg. Gap | Avg. Gap |
| *w/o* adaptive | | ✓ | ✓ | ✓ | 0.94% | 0.69‰ |
| *w/o* $\mathcal{L}_{\text{KD}}$ | ✓ | | ✓ | ✓ | 1.06% | 0.66‰ |
| *w/o* $\mathcal{L}_{\text{Task}}$ | ✓ | ✓ | | ✓ | 1.00% | 0.69‰ |
| off-policy | ✓ | ✓ | ✓ | | 10.42% | 0.80‰ |
| AMDKD | ✓ | ✓ | ✓ | ✓ | **0.90%** | **0.63‰** |

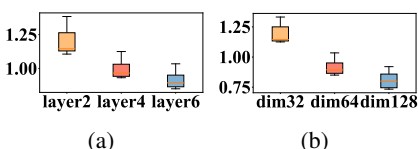

(a)         (b)

Figure 3: Effects of student network architectures. (a) different numbers of encoder layers, (b) different node embedding dimensions.

Table 5: Performance of AMDKD on CVRP-50 trained with different exemplar distributions.

| Model | Size (M) | Expansion Obj. | Gap | Implosion Obj. | Gap | Explosion Obj. | Gap | Grid Obj. | Gap | Uniform Obj. | Gap | Cluster Obj. | Gap | Mixed Obj. | Gap | Avg. Gap |
|---|---|---|---|---|---|---|---|---|---|---|---|---|---|---|---|---|
| LKH3 | - | 8.15 | - | 10.26 | - | 8.74 | - | 10.40 | - | 10.38 | - | 5.13 | - | 9.42 | - | - |
| POMO(Expansion) | 1.20 | 8.22 | 0.90% | 10.36 | 0.96% | 8.82 | 0.94% | 10.50 | 0.95% | 10.47 | 0.94% | 5.19 | 1.14% | 9.51 | 0.96% | 0.97% |
| POMO(Implosion) | 1.20 | 8.23 | 0.96% | 10.34 | 0.80% | 8.81 | 0.82% | 10.48 | 0.77% | 10.46 | 0.78% | 5.21 | 1.50% | 9.52 | 1.03% | 0.95% |
| POMO(Explosion) | 1.20 | 8.23 | 0.95% | 10.35 | 0.83% | 8.81 | 0.80% | 10.49 | 0.82% | 10.47 | 0.87% | 5.20 | 1.27% | 9.51 | 1.02% | 0.94% |
| POMO(Grid) | 1.20 | 8.23 | 0.97% | 10.34 | 0.79% | 8.81 | 0.82% | 10.48 | 0.76% | 10.46 | 0.77% | 5.21 | 1.57% | 9.52 | 1.07% | 0.97% |
| AMDKD-POMO* | 0.49 | 8.23 | 0.96% | 10.35 | 0.89% | 8.82 | 0.89% | 10.49 | 0.88% | 10.47 | 0.88% | 5.18 | 1.06% | 9.50 | 0.92% | **0.92%** |

lower than the 0.90% of AMDKD-POMO (64). However, this improvement comes at the expense of increased computational costs in three aspects: 1) more training time - AMDKD-POMO (128) needs almost 1.7 times (12 days vs 7 days) more training time; 2) more parameters - AMDKD-POMO (128) has about 2.4 times (1.20 M vs 0.49 M) more parameters; 3) slower inference time - AMDKD-POMO (128) infers about 1.2 times (1.3 min vs 1.1 min) slower than AMDKD-POMO (64). Hence, we note that there is a trade-off between solution quality and computational cost.

**Effects of different exemplar distributions.** We now vary the distributions which were used as the exemplar ones to validate that AMDKD could still be effective. We take the POMO and CVRP-50 as an example. To distinguish from the original version (AMDKD-POMO), we term the model taking Expansion, Implosion, Explosion, and Grid as the exemplar distributions as AMDKD-POMO*, and gathered the results in Table 5. It shows that the overall performance of AMDKD-POMO* is still better than POMO trained on these four distributions respectively. Hence, our AMDKD is able to consistently improve the cross-distribution generalization with different exemplar distributions.

## 6   Conclusions and future work

In this paper, we propose an Adaptive Multi-Distribution Knowledge Distillation (AMDKD) scheme to alleviate the cross-distribution generalization issue of deep models for VRPs. Different from the existing generalization methods, we aim to enhance the distribution generalization by transferring various policies learned from exemplar distributions into one via an efficient knowledge distillation scheme. To facilitate effective learning, we design an adaptive strategy to train a single yet generalist student network by leveraging multiple teachers in turns. The experiment results exhibit competitive performance of our AMDKD in generalizing to other unseen out-of-distribution instances (randomly generated or from benchmarks), which also consumes less computational resources. While our AMDKD is generic, a potential limitation is that its boost is not guaranteed to be always significant across all unseen distributions for all backbone models. For future work, we will investigate, 1) generalizing AMDKD for different/larger problem sizes; 2) considering the *improvement* models like DACT [14] as the backbone; 3) performing online distillation to jointly and efficiently train the teachers and the student models [55]; 4) assessing the impact of the quality of the validation dataset on the distillation; and 5) enhancing the interpretability of AMDKD [56].

## Acknowledgments and Disclosure of Funding

This work is supported in part by the National Key R&D Program of China (2018AAA0101203), the National Natural Science Foundation of China (62072483), and the Guangdong Basic and Applied Basic Research Foundation (2022A1515011690, 2021A1515012298); in part by the Agency for Science Technology and Research Career Development Fund (C222812027), and the IEO Decentralised GAP project of Continuous Last-Mile Logistics (CLML) at SIMTech (I22D1AG003).

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
