# Learning Generalizable Models for Vehicle Routing Problems via Knowledge Distillation (Appendix)

## A  Details of the considered distributions

In this paper, we consider various distributions for the node coordinates in VRPs, followed which we randomly generate instances for both training and testing. Below we present details on how to generate those instances. Specifically, we follow the recent work [10] and benchmark dataset TSPLIB [42] to generate instances of Uniform, Cluster and Mixed distributions, and follow the settings in [21, 43] for the Expansion, Implosion, Explosion, and Grid ones.

**Uniform distribution.** It considers uniformly distributed nodes. Following [10], we generate the two-dimensional coordinates $(x, y)$ of each node by sampling from a uniform space $\mathcal{U}([0, 1]^2)$. An exemplary instance is displayed in Figure 1(h).

**Cluster distribution.** It considers multiple ($n_c$) clusters, where we set $n_c = 3$. In specific, each cluster follows a normal distribution $\mathcal{N}(\mu, \sigma^2)^2$ with the mean sampled uniformly, i.e., $\mu \sim \mathcal{U}([0.2, 0.8]^2)$ and the standard deviation $\sigma = 0.07$. According to the $3\sigma$ rule, each node has a 99.7% probability of being generated in the $[0, 1]^2$ square region, and outliers will have their coordinates re-modified, where values less than 0 are changed to 0 and those greater than 1 are changed to 1, to ensure all coordinates are constrained to ($[0, 1]^2$). An exemplary instance is displayed in Figure 1(i).

**Mixed distribution.** It considers a mixture of the two distributions above, each with half of the nodes. For the latter, we only consider $n_c = 1$ cluster. An exemplary instance is displayed in Figure 1(j).

**Expansion distribution.** It considers a linear function to mutate the nodes in Uniform distribution. Gvien a randomly generated linear function $y = ax + b$, all nodes, orthogonal to the linear function within the distance $r$ ($r = 0.3$), are moved away from their original coordinates to a farther position, whose orthogonal distance is $r + \gamma$, where $\gamma$ obeys an exponential distribution with the rate parameter $\lambda = 10$, i.e. $\gamma \sim E(\lambda)$. Regarding the linear function, we first sample $b$ (intercept) from $[0, 1]$, then $a$ (slope) is sampled uniformly from $[0, 3]$ (if $b < 0.5$) or $[-3, 0]$ (if $b \geq 0.5$). Finally, we normalize all node coordinates $X$ as follows to ensure that they are constrained to $[0, 1]^2$,

$$X' = \frac{X - \min(X)}{\max(X) - \min(X)}, \tag{8}$$

where $X'$ denotes the normalized coordinates. An exemplary instance is displayed in Figure 1(k).

**Implosion distribution.** It considers an implosion to mutate the nodes in Uniform distribution. To simulate an implosion, it first samples a centroid $\epsilon_i$, then gathers all nodes within the circle of $\epsilon_i$ (with the radius $R_{ic} = 0.3$) together towards a new circle with the same centroid $\epsilon_i$ but a (randomly sampled) smaller radius ($R_i \leq 0.3$). An exemplary instance is displayed in Figure 1(l).

**Explosion distribution.** It considers to mutate the nodes in Uniform distribution by imitating the particles affected by an explosion. Similar to Implosion, it first randomly samples a centroid. Then, instead of gathering all nodes towards the centroid in Implosion distribution, it moves away those nodes from the circle (radius $R_{ec} = 0.3$) and explode them outside the circle, which follow the direction vector between the centroid $\epsilon_e$ and the corresponding nodes. The additive distance $\gamma$ is randomly sampled from an exponential distribution with a rate parameter, i.e. $\gamma \sim E(\lambda)$. All nodes are them normalized using Eq. (8). An exemplary instance is displayed in Figure 1(m).

**Grid distribution.** It considers to mutate the nodes in Uniform distribution by imposing a grid permutation. We first generate the four vertex of a square with the width and height equal to $R_g(R_g = 0.3)$ and then place it within the region $[0, 1]^2$. All the pre-generated nodes inside the box are re-arranged as a quadratic grid instead. In this case, all nodes are constrained to $[0, 1]^2$. An exemplary instance is displayed in Figure 1(n).

The above distributions are considered in both TSP and CVRP. For extra settings in CVRP, we follow the convention [10, 12, 14]. In specific, the demand $\delta_i$ of each node is sampled uniformly from $\mathcal{U}(1,$

$2, \cdots, 9$) and the capacity $Q$ of the vehicle varies with the problem scale, where we set $Q^{20} = 30$, $Q^{50} = 40$ and $Q^{100} = 50$. For instances from CVRPLIB, we exactly follow their settings.

## B Details of compared baselines

**Implementation Details.** We compare our AMDKD with various types of baselines. Regarding the neural baselines, we re-train LCP [13], HAC [17] and DACT [14] on our machine based on the code that are publicly available on Github. By default, we follow their original settings and the suggestion on the hyper-parameters. More details of the baselines are presented below.

- **Gurobi** [5]: we use Gurobi to obtain the optimal solutions to TSP instances, which are implemented under the default settings.
- **LKH** [4]: For CVRP, it is usually hard to obtain the optimal solutions. Thus, we use the strong LKH solver to find near-optimal solutions. Note that the LKH solver is also a widely used baseline to evaluate and compare the recent learning based methods for VRPs, where we run it following the conventions in [10, 12, 14, 23].
- **LCP** [13]: LCP is a two-stage method, where a *seeder* generates diverse initial solutions and a *reviser* rewrites the current solutions partially. We re-train the LCP on a mixed dataset containing instances from the three exemplar distributions. For inference of TSP-50 and TSP-100, we employ the LCP* (the best version reported in [13]) with two revisers (the lengths of the tour for revision are set to $\ell_{r1}$=10 and $\ell_{r2}$=20) and a sampling strategy (1,280), and set the total number of revision iteration $T_r$ to 45 (i.e., $T_{r1}$=25, $T_{r2}$=20, respectively). For TSP-20, we exploit one reviser ($\ell_r$=10) since the revision length must be less than the problem size according to its design, and set the number of revision iteration $T_r$ to 10.
- **HAC** [17]: HAC designs a hardness-adaptive Gaussian instances generator to produce instances to fine-tune the given pre-trained AM model. In its original design, the dataset used for fine-tuning contains half instances uniformly distributed and the other half produced by its own generator. In this paper, we substitute the instances of uniform distribution with a mixed dataset containing instances from the three exemplar distributions.
- **DACT** [14]: DACT learns to guide the pairwise operator to perform local search. We adopt the 2-opt version since it reports the best result for TSP and CVRP according to its original paper [14]. We re-train DACT on a mixed dataset containing instances from the three exemplar distributions, and set the iteration number to 1,280 for inference.

**Effects of distribution mixture augmentation.** To ensure fair comparisons, we re-train the baselines on a mixed dataset containing instances from the three exemplar distributions (with $^{\#}$). However, we notice that this simple distribution augmentation does not always lead to a better generalization, espicially for the methods that have an *improvement* component. For example, regarding the *improvement* method DACT, we find that DACT$^{\#}$ performs even worse than DACT trained on Uniform distribution for CVRP-100; and regarding the hybrid method, LCP (U) performs slightly better than

Table 6: Effects of distribution mixture during training.

| | Model | Uniform | Cluster | Mixed | Grid | $n=50$ Implosion | Expansion | Explosion | Avg. | Uniform | Cluster | Mixed | Grid | $n=100$ Implosion | Expansion | Explosion | Avg. |
|---|---|---|---|---|---|---|---|---|---|---|---|---|---|---|---|---|---|
| TSP | Gurobi | 5.70 | 2.65 | 4.92 | 5.69 | 5.60 | 4.38 | 4.62 | - | 7.76 | 3.66 | 6.73 | 7.79 | 7.61 | 5.39 | 5.83 | - |
| | POMO (U) | 5.70 | 2.66 | 4.93 | 5.69 | 5.60 | 4.38 | 4.62 | 0.12% | 7.78 | 3.73 | 6.79 | 7.8 | 7.62 | 5.43 | 5.85 | 0.61% |
| | POMO$^{\#}$ | 5.70 | 2.66 | 4.93 | 5.69 | 5.60 | 4.38 | 4.62 | **0.06%** | 7.78 | 3.67 | 6.75 | 7.81 | 7.63 | 5.42 | 5.84 | **0.29%** |
| | AM (U) | 5.73 | 2.71 | 4.99 | 5.73 | 5.63 | 4.43 | 4.65 | **1.02%** | 7.93 | 3.93 | 7.00 | 7.95 | 7.78 | 5.64 | 5.98 | 3.58% |
| | AM$^{\#}$ | 5.73 | 2.72 | 4.99 | 5.73 | 5.64 | 4.43 | 4.65 | 1.06% | 7.92 | 3.90 | 7.00 | 7.95 | 7.77 | 5.66 | 5.98 | **3.40%** |
| | LCP (U) | 5.72 | 3.58 | 5.01 | 5.71 | 5.62 | 4.44 | 4.65 | 5.70% | 7.95 | 6.91 | 7.34 | 7.98 | 7.81 | 6.23 | 6.26 | **18.31%** |
| | LCP$^{\#}$ | 5.73 | 3.11 | 5.04 | 5.72 | 5.63 | 4.47 | 4.65 | **3.44%** | 7.99 | 6.81 | 7.34 | 8.02 | 7.85 | 6.41 | 6.30 | 18.72% |
| | HAC (U) | 5.81 | 2.87 | 5.07 | 5.80 | 5.71 | 4.48 | 4.70 | 3.00% | 8.14 | 4.24 | 7.26 | 8.17 | 7.99 | 5.96 | 6.16 | 7.78% |
| | HAC$^{\#}$ | 5.72 | 2.70 | 4.97 | 5.72 | 5.63 | 4.41 | 4.64 | **0.78%** | 7.96 | 3.94 | 6.98 | 7.98 | 7.81 | 5.60 | 6.01 | **3.70%** |
| | DACT (U) | 5.70 | 2.69 | 4.98 | 5.70 | 5.61 | 4.43 | 4.63 | 0.61% | 7.90 | 3.90 | 6.98 | 7.92 | 7.75 | 5.80 | 5.99 | 3.67% |
| | DACT$^{\#}$ | 5.71 | 2.66 | 4.94 | 5.71 | 5.62 | 4.40 | 4.63 | **0.33%** | 7.96 | 3.74 | 6.88 | 7.98 | 7.80 | 5.67 | 5.98 | **2.80%** |
| CVRP | LKH3 | 10.38 | 5.13 | 9.42 | 10.40 | 10.26 | 8.15 | 8.74 | - | 15.65 | 7.81 | 14.19 | 15.64 | 15.44 | 11.39 | 12.32 | - |
| | POMO (U) | 10.46 | 5.21 | 9.52 | 10.49 | 10.35 | 8.23 | 8.81 | 0.98% | 15.80 | 7.99 | 14.38 | 15.87 | 15.59 | 11.55 | 12.45 | 1.36% |
| | POMO$^{\#}$ | 10.47 | 5.18 | 9.50 | 10.50 | 10.36 | 8.23 | 8.82 | **0.93%** | 15.83 | 7.91 | 14.32 | 15.82 | 15.62 | 11.55 | 12.46 | **1.18%** |
| | AM (U) | 10.64 | 5.35 | 9.70 | 10.66 | 10.52 | 8.39 | 8.96 | **2.92%** | 16.13 | 8.58 | 14.84 | 16.13 | 15.93 | 11.93 | 12.77 | 4.59% |
| | AM$^{\#}$ | 10.63 | 5.37 | 9.71 | 10.66 | 10.52 | 8.40 | 8.97 | 2.97% | 16.16 | 8.46 | 14.85 | 16.15 | 15.95 | 11.93 | 12.78 | **4.47%** |
| | DACT (U) | 10.54 | 5.24 | 9.57 | 10.56 | 10.42 | 8.26 | 8.88 | 1.62% | 16.15 | 8.17 | 14.74 | 16.14 | 15.94 | 12.33 | 12.74 | **4.22%** |
| | DACT$^{\#}$ | 10.54 | 5.21 | 9.57 | 10.57 | 10.43 | 8.27 | 8.89 | **1.59%** | 16.52 | 8.25 | 14.97 | 16.51 | 16.30 | 12.02 | 13.01 | 5.56% |

LCP$^{\#}$ on TSP-100. Nevertheless, this distribution augmentation improves the generalization of construction methods POMO [12] and AM [10] on larger instances, i.e., TSP-100 and CVRP-100.

**Why AMDKD could be better than distribution mixture augmentation?** Recall that we conclude from Table 2 that our AMDKD usually achieves better cross-distribution generalization performance when compared to the above distribution mixture method (# models). Though this still remains a open question, we list some possible intuitions on why AMDKD works better as follow:

- Different exemplar distributions may have different levels of difficulty for solving. Without additional intervention, reinforcement learning tends to learn those easier things to get a higher reward. This means that training on mixed data may possibly bias the learning towards distributions that are easier to solve (a "winner-take-all" issue). On the contrary, our AMDKD selects a specific teacher model based on the weakness of the current student model (by our adaptive strategy), which encourages it to learn those hard-to-solve distributions. And this adaptive strategy echoes how humans learn knowledge, where more time is always required for harder subjects.

- Directly training on a mixed dataset may not be efficient or stable. On one hand, it would be hard for deep reinforcement learning to directly learn good patterns from mixed data that follow different distributions, due to the possibly limited representation capability of the neural networks for handling such hard optimization problems, even without the diversity in distributions. On the other hand, the tasks for different distributions may have different reward ranges (such as the route length), which may cause instability in the RL training. For example, the average total rewards for solving CVRP-100 instances following Uniform and Cluster distributions are around 16 and 7, respectively. Our AMDKD tackles this issue in a way that only one distribution is leveraged in a training epoch. Further empowered by the knowledge distillation, our AMDKD framework efficiently and effectively transfers useful knowledge (patterns) from various teacher models to a unified and light student model.

## C  Additional analysis of AMDKD

### C.1  Effects of multiple teacher co-training.

Recall that AMDKD selects only one distribution and its corresponding teacher in each epoch. Different from ours, some existing multi-teacher knowledge distillation approaches exploit multiple teachers simultaneously in each epoch. We term such strategy as MT, whose loss function follows Eq. (4). In Figure 4, we draw the boxplot of the overall gaps (on all seven test distributions for CVRP-50) of AMDKD-POMO and its MT version, and compare the results using the Wilcoxon test. As clearly demonstrated, allowing unprofessional teachers to advise for distributions in which they are not specialized will interfere with the process of knowledge distillation, inducing significantly inferior performance compared to ours.

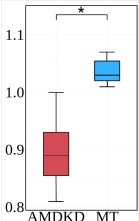

Figure 4: Boxplot of the overall gaps of AMDKD and its MT version. Here, $\star$ in the plot means that the two models are much different with statistical significance $p$-value $= 0.02 < 0.05$ (Wilcoxon test).

### C.2  Effects of validation datasets.

Recall that the adaptive probability of teacher selection in Eq. (5) is calculated by the real-time performance of student model on the validation datasets, we further investigate whether the size of the validation datasets $\mathcal{V}$ will largely influence AMDKD. As displayed in Figure 5 and Figure 6, increasing the size of $\mathcal{V}$ (from 1,000 to 2,000) slightly improves the performance of AMDKD

(but with no statistical significance), whereas decreasing the size of $\mathcal{V}$ (from 1,000 to 500) slightly impairs the performance of AMDKD (also with no statistical significance), which indicates that the good performance of AMDKD may not rely on the size of the validation datasets. Meanwhile, performing student model evaluation in each epoch inevitably introduces additional computation cost, where larger size of the validation datasets will cause longer training time. However, we note that the increment of validation in training time is acceptable when we use $\mathcal{V} = 1,000$. Taking AMDKD-AM training on CVRP-100 as an example, the total evaluation time is 2.7s (0.9s per exemplar distribution on average) for each epoch, which is approximately 1% of the total training time (i.e., 4 min). What's more, we note that the likelihood of selecting different teacher models would eventually converge to a stable one, which means that we may stop such evaluation early to speed up the training further if needed. For example, for training AMDKD-AM on CVRP-100 (see Figure 7), we may stop the student evaluation early at around 3,000 epochs.

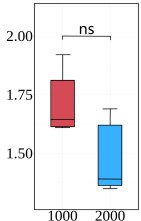

Figure 5: Boxplot of the overall gaps of AMDKD-AM (with size of $\mathcal{V}$=1,000, red) and AMDKD-AM (with size of $\mathcal{V}$=2,000, blue) on CVRP-50. The "ns" in the plot means that the the two models are not sigificantly different with statistical significance $p$-value $= 0.25 > 0.1$ (Wilcoxon test).

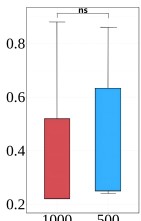

Figure 6: Boxplot of the overall gaps of AMDKD-AM (with size of $\mathcal{V}$=1,000, red) and AMDKD-AM (with size of $\mathcal{V}$=500, blue) on TSP-50. The "ns" in the plot means that the two models are not sigificantly different with statistical significance $p$-value $= 0.36 > 0.1$ (Wilcoxon test).

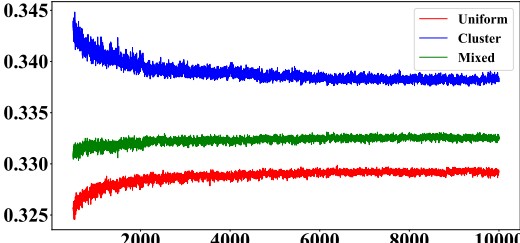

Figure 7: Likelihood of teacher selection along the epochs (AMDKD-AM on CVRP-100).

## C.3 Effects of different hyper-parameters.

We further discuss the influence of the hyper-parameters on the performance of AMDKD.

- The starting epoch of the adaptive teacher selection strategy ($E'$): it indicates the epoch to start our adaptive strategy for teacher selection. We include this hyper-parameter because preliminary experiments revealed that there could be a point in the learning curve (i.e., the reward convergence curve) where the training curves without and with (starting at the first epoch $E'$=1) the adaptive strategy may meet. This suggests that there might be a sweet spot

to implement the proposed adaptive strategy. For AMDKD-AM, the sweet spot is around $E'$=500. And for AMDKD-POMO, we do not observe such a pattern and thus we use $E'$)=1. In figure 8, we provide an example of how $E'$ will affect the performance of AMDKD-AM.

- Number of steps per epoch ($T$): it indicates how long will the student model learn from the selected teacher before it possibly switches to a new one, which should not be too small or too large. As for POMO, the original $T$ is about 20, and we did not change it. As for AM, the original $T$ is about 2,500, and we empirically reduce it to 250 for a better trade-off.

- The total number of training epochs ($E$): it mainly follows the settings of the original backbone. In this paper, we employ different training epochs for different sizes and tasks since the hardness of the task itself may grow with size, which may require more steps to converge. Training curves of AMDKD-POMO (for CVRP) are depicted in Figure 9.

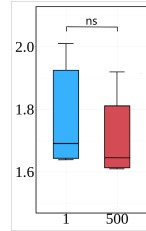

Figure 8: Boxplot of the overall gaps of AMDKD-AM ($E'$=500, red) and AMDKD-AM ($E'$=1, blue) on CVRP-50. The "ns" in the plot means that the two models are not significantly different with statistical significance $p$-value = $0.37 > 0.1$ (Wilcoxon test).

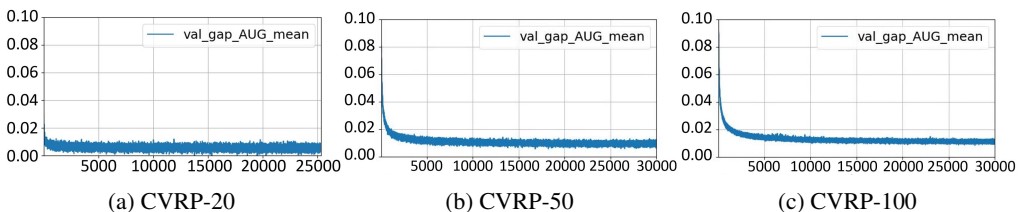

(a) CVRP-20      (b) CVRP-50      (c) CVRP-100

Figure 9: Training curves of AMDKD-POMO for solving CVRP. The x-axis is the epoch and the y-axis is the average gaps on the used three exemplar distributions.

## C.4 Stability studies of AMDKD.

To demonstrate the stability of our experiment results, we take CVRP-50 as an example and independently run our trained AMDKD-AM and AM$^{\#}$ for 10 times, where we adopt different random seeds during the sampling process. As shown in Figure 10, both AMDKD-AM and AM$^{\#}$ exhibit extremely small fluctuations (even less than 0.001) when running with different seeds. Based on the Wilcoxon test, our AMDKD-AM significantly (with $p$-value < 0.001) outperforms AM$^{\#}$ on every unseen in-distributions and OoD distributions. As for AMDKD-POMO and POMO$^{\#}$, the greedy decoding strategy is adopted, hence the results are expected to be stable with almost no fluctuation.

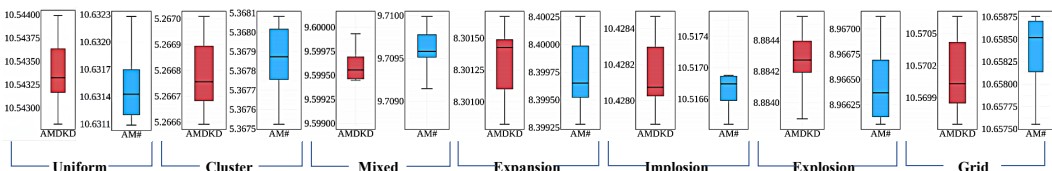

Figure 10: Experiment results of AMDKD-AM (red) and AM$^{\#}$ (blue) with different random seeds.

## D   Detailed results on benchmark datasets

We present the detailed results of benchmark dataset in Table 8 (TSPLIB) and Table 9 (CVRPLIB), respectively. As displayed, models that have only been trained on the uniform distribution, i.e., AM and POMO, perform extremely poorly when inferring instances that may follow unknown distributions from the benchmark. The upgraded POMO$^{\#}$ significantly outperforms POMO, which seemingly alleviates this issue through training on our exemplar distributions, however, this simple strategy does not work well with AM. Regarding the prior cross-distribution generalization methods including GANCO, HAC, PSRO and DROP which show the potential to improve AM or POMO, their results are still far from satisfactory, where the performance of DROP are even inferior to POMO$^{\#}$ trained on our exemplar distributions. Nevertheless, our AMDKD not only outperforms all these baselines, but also brings a much more significant improvement over the backbone AM and POMO than those baselines do. Meanwhile, our AMDKD-POMO exhibits a much better generalization over POMO$^{\#}$ on CVRPLIB. Finally, we note that AMDKD equipped with EAS performs the best among all baselines, even achieving the optimal solution on some instances (e.g., KroA100, KroD100, lin105 in TSPLIB and X-n110-k14 in CVRPLIB). This leads to a new state-of-the-art performance for neural methods on these benchmark datasets. Finally, we list the full generalization results of our AMDKD in Table 10 (TSPLIB) and Table 11 (CVRPLIB), respectively.

Table 8: Detailed generalization results on selected instances from TSPLIB.

| Instance | Opt. | PSRO | AM | GANCO | HAC | AM$^{\#}$ | AMDKD-AM | POMO | DROP | POMO$^{\#}$ | AMDKD-POMO | AMDKD+EAS |
|---|---|---|---|---|---|---|---|---|---|---|---|---|
| KroA100 | 21282 | 21703 | 46621 | 21908 | 21838 | 22138 | 21650 | 38452 | 24623 | 21285 | 21285 | 21282 |
| KroB100 | 22141 | 22855 | 37921 | 22956 | 23110 | 23189 | 22350 | 33521 | 24874 | 22197 | 22233 | 22195 |
| KroC100 | 20749 | 21079 | 34258 | 21139 | 21068 | 22326 | 21279 | 30736 | 24785 | 20751 | 20752 | 20947 |
| KroD100 | 21294 | 21828 | 36141 | 21929 | 22625 | 23093 | 21863 | 29512 | 23257 | 21352 | 21314 | 21294 |
| KroE100 | 22068 | 22532 | 29628 | 23174 | 22807 | 22865 | 22327 | 26829 | 26057 | 22179 | 22185 | 22111 |
| lin105 | 14379 | 15372 | 15148 | 15478 | 15003 | 16865 | 14988 | 14922 | 14688 | 14430 | 14430 | 14379 |
| pr107 | 44303 | 45288 | 53846 | 45393 | 47250 | 76152 | 46146 | 52846 | 47853 | 44647 | 45022 | 44347 |
| Avg. Gap ($n$=100-150) | 0.00% | 2.93% | 55.19% | 3.80% | 4.16% | 16.80% | 2.50% | 37.63% | 12.14% | 0.31% | 0.44% | 0.21% |
| ch150 | 6528 | 6866 | 6930 | 6704 | 6852 | 6680 | 6669 | 6844 | 6709 | 6574 | 6609 | 6554 |
| rat195 | 2323 | 2600 | 2612 | 2585 | 2638 | 3237 | 2550 | 2554 | 2403 | 2422 | 2432 | 2406 |
| kroA200 | 29368 | 31450 | 35637 | 31741 | 33174 | 34294 | 31112 | 34972 | 34275 | 29840 | 29906 | 29931 |
| Avg. Gap ($n$=150-200) | 0.00% | 8.06% | 13.32% | 7.36% | 10.50% | 19.48% | 5.96% | 11.29% | 7.64% | 2.19% | 2.59% | 1.96% |

Table 9: Detailed generalization results on selected instances from CVRPLIB.

| Instance | Opt. | AM | AM$^{\#}$ | AMDKD-AM | POMO | DROP | POMO$^{\#}$ | AMDKD-POMO | AMDKD+EAS |
|---|---|---|---|---|---|---|---|---|---|
| X-n101-k25 | 27591 | 38264 | 30327 | 30782 | 29484 | 28949 | 30510 | 29299 | 27855 |
| X-n106-k14 | 26362 | 27923 | 27958 | 27279 | 27762 | 27308 | 27077 | 26847 | 26550 |
| X-n110-k13 | 14971 | 16320 | 15668 | 15348 | 15896 | 15386 | 15175 | 15315 | 14971 |
| X-n115-k10 | 12747 | 14055 | 14638 | 13366 | 13952 | 13783 | 13609 | 13418 | 12883 |
| X-n120-k6 | 13332 | 14456 | 16094 | 14162 | 14351 | 14058 | 13997 | 13604 | 13457 |
| X-n125-k30 | 55539 | 74329 | 68870 | 58507 | 69560 | 61382 | 62383 | 58570 | 56596 |
| X-n129-k18 | 28940 | 30869 | 30833 | 29851 | 30155 | 30075 | 29597 | 29449 | 29007 |
| X-n134-k13 | 10916 | 13952 | 12709 | 12573 | 13483 | 12846 | 11325 | 11330 | 11073 |
| X-n139-k10 | 13590 | 14893 | 14953 | 14097 | 14132 | 13979 | 14053 | 13955 | 13704 |
| X-n143-k7 | 15700 | 18251 | 18345 | 16509 | 17923 | 17682 | 16487 | 16346 | 15871 |
| Avg. Gap ($n$ 100-150) | 0.00% | 16.65% | 13.00% | 6.12% | 10.66% | 7.25% | 5.32% | 3.54% | 0.92% |
| X-n153-k22 | 21220 | 38423 | 24722 | 23766 | 26386 | 24386 | 23629 | 23590 | 21849 |
| X-n157-k13 | 16876 | 22051 | 19890 | 17539 | 19978 | 18378 | 17950 | 17450 | 17093 |
| X-n181-k23 | 25569 | 27826 | 27314 | 26415 | 27428 | 27094 | 29014 | 26756 | 25736 |
| X-n190-k8 | 16980 | 37820 | 21020 | 21162 | 22310 | 19864 | 18912 | 17575 | 17228 |
| X-n200-k36 | 58578 | 76528 | 66298 | 62335 | 73135 | 64921 | 62228 | 62967 | 60562 |
| Avg. Gap ($n$ =150-200) | 0.00% | 54.79% | 15.63% | 10.06% | 21.25% | 11.52% | 9.76% | 6.04% | 1.95% |

## E   Used assets and licenses

Table 12 lists the used assets in our work, which are all open-source for academic research. For our code and used data (new assets), we are using the MIT License.

Table 10: Full generalization results on TSPLIB (instances ranged from 100 to 200).

| Instance | Opt. | AM# | | AMDKD-AM | | POMO# | | AMDKD-POMO | | AMDKD+EAS | |
|---|---|---|---|---|---|---|---|---|---|---|---|
| | | Obj. | Gap | Obj. | Gap | Obj. | Gap | Obj. | Gap | Obj. | Gap |
| kroA100 | 21282 | 22138 | 4.02% | 21650 | 1.73% | 21285 | 0.02% | 21285 | 0.02% | 21282 | 0.00% |
| kroB100 | 22141 | 23189 | 4.73% | 22350 | 0.94% | 22197 | 0.25% | 22233 | 0.41% | 22195 | 0.24% |
| kroC100 | 20749 | 22326 | 7.60% | 21279 | 2.55% | 20751 | 0.01% | 20752 | 0.02% | 20947 | 0.95% |
| kroD100 | 21294 | 23093 | 8.45% | 21863 | 2.67% | 21352 | 0.27% | 21314 | 0.09% | 21294 | 0.00% |
| kroE100 | 22068 | 22865 | 3.61% | 22327 | 1.17% | 22179 | 0.50% | 22185 | 0.53% | 22111 | 0.19% |
| eil101 | 629 | 663 | 5.45% | 647 | 2.82% | 641 | 1.90% | 645 | 2.55% | 629 | 0.00% |
| lin105 | 14379 | 16865 | 17.29% | 14988 | 4.24% | 14430 | 0.36% | 14430 | 0.36% | 14379 | 0.00% |
| pr107 | 44303 | 76152 | 71.89% | 46146 | 4.16% | 44647 | 0.78% | 45022 | 1.62% | 44347 | 0.10% |
| pr124 | 59030 | 62075 | 5.16% | 60042 | 1.71% | 59031 | 0.00% | 59281 | 0.43% | 59030 | 0.00% |
| bier127 | 118282 | 275748 | 133.13% | 123211 | 4.17% | 119232 | 0.80% | 119052 | 0.65% | 118729 | 0.38% |
| ch130 | 6110 | 6231 | 1.98% | 6171 | 1.00% | 6146 | 0.60% | 6152 | 0.69% | 6115 | 0.08% |
| pr136 | 96772 | 100194 | 3.54% | 99912 | 3.24% | 98478 | 1.76% | 98215 | 1.49% | 97487 | 0.74% |
| pr144 | 58537 | 66628 | 13.82% | 60807 | 3.88% | 59034 | 0.85% | 58956 | 0.72% | 58794 | 0.44% |
| ch150 | 6528 | 6680 | 2.33% | 6669 | 2.16% | 6574 | 0.70% | 6609 | 1.25% | 6554 | 0.40% |
| kroA150 | 26524 | 29501 | 11.22% | 27354 | 3.13% | 26723 | 0.75% | 26808 | 1.07% | 26538 | 0.05% |
| kroB150 | 26130 | 28585 | 9.39% | 26820 | 2.64% | 26334 | 0.78% | 26328 | 0.76% | 26152 | 0.08% |
| pr152 | 73682 | 85704 | 16.31% | 78120 | 6.02% | 74673 | 1.35% | 75270 | 2.16% | 75250 | 2.13% |
| rat195 | 2323 | 3237 | 39.35% | 2550 | 9.77% | 2422 | 4.25% | 2432 | 4.68% | 2406 | 3.57% |
| kroA200 | 29368 | 34294 | 16.77% | 31112 | 5.94% | 29840 | 1.61% | 29906 | 1.83% | 29931 | 1.92% |
| kroB200 | 29437 | 34074 | 15.75% | 31968 | 8.60% | 29665 | 0.77% | 30132 | 2.36% | 29765 | 1.11% |
| Avg. Gap | | | 19.59% | | 3.63% | | 0.92% | | 1.18% | | 0.62% |

Table 11: Full generalization results on CVRPLIB (instances ranged from 100 to 200).

| Instance | Opt. | AM# | | AMDKD-AM | | POMO# | | AMDKD-POMO | | AMDKD+EAS | |
|---|---|---|---|---|---|---|---|---|---|---|---|
| | | Obj. | Gap | Obj. | Gap | Obj. | Gap | Obj. | Gap | Obj. | Gap |
| X-n101-k25 | 27591 | 30327 | 9.92% | 30782 | 11.57% | 30510 | 10.58% | 29299 | 6.19% | 27855 | 0.96% |
| X-n106-k14 | 26362 | 27958 | 6.06% | 27279 | 3.48% | 27077 | 2.71% | 26847 | 1.84% | 26550 | 0.71% |
| X-n110-k13 | 14971 | 15668 | 4.66% | 15348 | 2.52% | 15175 | 1.36% | 15315 | 2.30% | 14971 | 0.00% |
| X-n115-k10 | 12747 | 14638 | 14.83% | 13366 | 4.86% | 13609 | 6.76% | 13418 | 5.27% | 12883 | 1.07% |
| X-n120-k6 | 13332 | 16094 | 20.71% | 14162 | 6.23% | 13997 | 4.99% | 13604 | 2.04% | 13457 | 0.94% |
| X-n125-k30 | 55539 | 68870 | 24.00% | 58507 | 5.34% | 62383 | 12.32% | 58570 | 5.46% | 56596 | 1.90% |
| X-n129-k18 | 28940 | 30833 | 6.54% | 29851 | 3.15% | 29597 | 2.27% | 29449 | 1.76% | 29007 | 0.23% |
| X-n134-k13 | 10916 | 12709 | 16.43% | 12573 | 15.18% | 11325 | 3.74% | 11330 | 3.79% | 11073 | 1.44% |
| X-n139-k10 | 13590 | 14953 | 10.03% | 14097 | 3.73% | 14053 | 3.41% | 13955 | 2.69% | 13704 | 0.84% |
| X-n143-k7 | 15700 | 18345 | 16.84% | 16509 | 5.15% | 16487 | 5.01% | 16346 | 4.12% | 15871 | 1.09% |
| X-n148-k46 | 43448 | 61800 | 42.24% | 52627 | 21.13% | 53217 | 22.48% | 46993 | 8.16% | 44075 | 1.44% |
| X-n153-k22 | 21220 | 24722 | 16.50% | 23766 | 12.00% | 23629 | 11.35% | 23590 | 11.17% | 21849 | 2.96% |
| X-n157-k13 | 16876 | 19890 | 17.86% | 17539 | 3.93% | 17950 | 6.36% | 17450 | 3.40% | 17093 | 1.29% |
| X-n162-k11 | 14138 | 14762 | 4.41% | 14663 | 3.72% | 14951 | 5.75% | 14903 | 5.41% | 14543 | 2.86% |
| X-n167-k10 | 20557 | 21686 | 5.49% | 21468 | 4.43% | 21573 | 4.94% | 21401 | 4.11% | 20890 | 1.62% |
| X-n172-k51 | 45607 | 62419 | 36.86% | 64444 | 41.30% | 49844 | 9.29% | 49741 | 9.06% | 46340 | 1.61% |
| X-n176-k26 | 47812 | 53263 | 11.40% | 51102 | 6.88% | 54149 | 13.25% | 53189 | 11.25% | 49241 | 2.99% |
| X-n181-k23 | 25569 | 27314 | 6.83% | 26415 | 3.31% | 29014 | 13.47% | 26756 | 4.64% | 25736 | 0.65% |
| X-n186-k15 | 24145 | 25845 | 7.04% | 25526 | 5.72% | 25827 | 6.97% | 26332 | 9.06% | 24893 | 3.10% |
| X-n190-k8 | 16980 | 21020 | 23.79% | 21162 | 24.63% | 18912 | 11.38% | 17575 | 3.50% | 17228 | 1.46% |
| X-n195-k51 | 44225 | 57830 | 30.76% | 60882 | 37.66% | 48907 | 10.59% | 51284 | 15.96% | 45758 | 3.47% |
| Avg. Gap | | | 15.87% | | 10.76% | | 8.05% | | 5.77% | | 1.55% |

Table 12: Used assets and their licenses.

| Type | Asset | License | Usage |
|---|---|---|---|
| Code | Gurobi [5] | Free Academic lisence | Evaluation |
| | LKH3 [4] | Available for academic use | Evaluation |
| | AM [10] | MIT License | Remodification and evaluation |
| | POMO [12] | MIT License | Remodification and evaluation |
| | LCP [13] | MIT License | Remodification and evaluation |
| | HAC [17] | MIT License | Remodification and evaluation |
| | EAS [23] | MIT License | Remodification and evaluation |
| | DACT [14] | MIT License | Remodification and evaluation |
| | tspgen [43] | GNU General Public License v3.0 | Generating datasets |
| Datasets | TSPLIB [42] | Available for any non-commerial use | Testing |
| | CVRPLIB [22] | Available for any non-commerial use | Testing |