# OpenReview forum: "Learning Generalizable Models for Vehicle Routing Problems via Knowledge Distillation"
_NeurIPS.cc/2022/Conference — NeurIPS 2022 Accept_

### Official Review · Reviewer_iBvW · 2022-07-12

**Rating:** 5
**Confidence:** 1
**Soundness:** 3 good
**Presentation:** 3 good
**Contribution:** 3 good

**Summary:**

This paper aims to enhance the cross-distribution generalizability of deep learning models for the vehicle routing problem. It proposes a novel method that is based on knowledge distillation and uses several teacher networks to guide learning a student network that is both light-weight and of strong generalizability. Moreover, it proposes an adaptive strategy to select the teacher models that are more informative. Experiments on two datasets verify the effectiveness of the proposed method by adding it on two existing methods and achieves performance gains.

**Questions:**

See the weakness above.

**Ethics Review Area:**

["I don’t know"]

**Limitations:**

Limitations are well addressed.

**Strengths And Weaknesses:**

Strengths
1) The proposed method is technically sound.
2) Experiments on two datasets verify the effectiveness of the proposed method by comparing with existing methods.
3) The proposed method is overall well presented.


Weaknesses:
1) From the implementation details, it seems that the proposed method has quite a few task-specific designs, e.g., training policies, architectures, hyper-parameters, etc. It is thus doubtful if the proposed method is generalizable enough when applied in practice.

2) In Table 1, there are many 0.00%, especially for methods LKH and Gurobi, all their result numbers are 0.00%. I am not sure if it has some special implications. If it is, there should be some explanations about it; otherwise, it could be presented in a better way, or maybe just leave it unfilled.

3) What is the performance variance with respect to the hyper-parameters.

---

> ### Author Response · Authors · 2022-08-02
> **Response to Reviewer #iBvW (continued)**
>
> **[Regarding the performance variance w.r.t. hyper-parameters]**
>
> In our original submission, we have already included a series of ablation studies on how good AMDKD is under different hyper-parameter settings (Section 5.3). For example, pertaining to the student network architectures, we have explored two hyper-parameters (the embedding dimensions of the entire network and the number of layers in its encoder) of the student network architectures in our paper. As shown in Fig. 3, generally,  the larger the student model, the better its solution quality. In light of the trade-off between computation cost and solution quality, we eventually curtail the dimension of the node embeddings from 128 (teacher) to 64 (student). As for other hyper-parameters (e.g., E’ and T), we have demonstrated that they are somehow easy to choose. We do acknowledge that it is meaningful to further conduct more experiments on hyper-parameters to specifically analyze their influence on the eventual performance, which we will include in the revised manuscript (it may need a couple of weeks to implement and run the new experiments).
>
> ---
> **Reference**
> ```
> [11] Wouter Kool, Herke van Hoof, and Max Welling. Attention, learn to solve routing problems! In International Conference on Learning Representations, 2018.
> [13] Yeong-Dae Kwon, Jinho Choo, Byoungjip Kim, Iljoo Yoon, Youngjune Gwon, and Seungjai Min. POMO: Policy optimization with multiple optima for reinforcement learning. In Advances in Neural Information Processing Systems, volume 33, pages 21188–21198, 2020.
> [14] Minsu Kim, Jinkyoo Park, and Joungho Kim. Learning collaborative policies to solve np-hard routing problems. In Advances in Neural Information Processing Systems, volume 34, pages 10418–10430, 2021.
> ```

---

> > ### Comment · Reviewer_iBvW · 2022-08-09
> > **Keep my score.**
> >
> > Thanks for the reply. I am not familiar with topic, but the paper and authors' response sound reasonable. Also, other reviewers are positive to this work. So, I will keep my positive rating as well.

---

> ### Author Response · Authors · 2022-08-02
> **Response to Reviewer #iBvW**
>
> We appreciate the reviewer for the positive and insightful feedback. We are delighted that the reviewer acknowledged the soundness of our work. The concerns from the reviewer are responded as follows.
>
> ---
> **[Regarding the task-specific designs in AMDKD]**
>
> For the task-specific hyper-parameters of the backbone models (i.e., AM and POMO) like training policies and architectures, almost all of them are set directly according to the suggested values/designs in their original papers and we did not change them in our experiments. For our proposed AMDKD (algorithm 1), it involves only a few new hyper-parameters, which are easy to choose in practice. Therefore, we believe that our AMDKD is generic enough for new tasks as long as the backbone neural model can solve the problem well on its original distribution. And as also acknowledged by reviewer *#kcQQ*, our AMDKD is a generic framework that could be easily applied to existing neural methods. Below we explain more details/guidelines on choosing these major hyper-parameters.
>
> - Training policies: For boosting both AM and POMO via AMDKD to solve TSP and CVRP, we employ exactly the same design of the original setting including the RL algorithm, the MDP (e.g., state, action, reward), etc.
>
> - Network architectures: We directly use the same architectures of the original AM and POMO as teacher models and we propose to reduce the dimension of the policy network to be half for the student model. We have already verified this design in Section 5.3 (see our response to the last comment below).
>
> - Other hyper-parameters in AMDKD:
>
>
>   1) The starting epoch of the adaptive teacher selection strategy (E’): it indicates the epoch to start our adaptive strategy for teacher selection. We include this hyperparameter because preliminary experiments revealed that there could be a point in the learning curve (i.e., the reward convergence curve) where the training curves without and with (starting at the first epoch E’=1) the adaptive strategy may meet. This suggests that there might be a sweet spot to implement the proposed adaptive strategy. For AMDKD-AM, the sweet spot is around E’=500. And for AMDKD-POMO, we do not observe such a pattern and thus we use E’=1.
>
>   2) Number of steps per epoch (T): it indicates how long will the student model learn from the selected teacher before it possibly switches to a new one, which should not be too small or too large. As for POMO, the original T is about 20, and we did not change it. As for AM, the original T is about 2500, and we empirically reduce it to 250 for a better trade-off.
>
>   3) The total number of training epochs (E): it mainly follows the settings of the original backbone. In this paper, we employ different training epochs for different sizes and tasks since the hardness of the task itself may grow with size, which may require more steps to converge.
>
> Furthermore, we have already done a series of ablation studies (see Section 5.3 and Appendix C.1) to verify that our AMDKD is effective in solving different tasks (e.g., TSP and CVRP experiments in Table 4). We will add more discussions/guidelines on how to select the major hyper-parameters in the revised manuscript.
>
> ---
> **[Regarding the 0.00% (s) in Table2]**
>
> We would like to explain that this kind of presentation is commonly adopted in the literature on neural combinatorial optimization (e.g., [11][14]). As explained in Line 244, the numbers in Table 1 refer to the (optimality) gap between the corresponding method and the strong traditional solver LKH (for CVRP) and Gurobi (for TSP). Therefore, the results for the two solvers are all zero (0.00%). However, the 0.00% (s) in some other rows might not be all zeros (i.e., some are the rounded results of small non-zero values), which indicates that these methods perform well and could attain near-optimal solutions. We acknowledge that with the same presentation, these two kinds of results may have different implications. In light of the constructive comment and suggestions, we will present them in a better way in the revised paper. For example, fill the real zero ones (optimal results) with a hyphen as did in [13].

---

### Official Review · Reviewer_kcQQ · 2022-07-12

**Rating:** 6
**Confidence:** 3
**Soundness:** 3 good
**Presentation:** 3 good
**Contribution:** 3 good

**Summary:**

This paper proposes a knowledge distillation (KD) approach for learning generalizable models for vehicle route planning (VRP). One key challenge in VRP is identified, that is, cross-distribution generalization. The proposed KD approach (a.k.a. AMDKD) leverages multiple specialized teacher models from different distributions to distill a generalist student model that can handle multiple distributions at the same time and generalizes well to OOD instances. Overall, the paper is well-written, the proposed method is clearly explained, and the experiments demonstrate promising results of the hybrid distillation scheme for VRP.


**Questions:**

1. Table 2 shows the time* of AMDKD+EAS methods is much slower than existing methods, but the authors claimed that the proposed approach can help attain a more efficient student model. Why is that?
2. Maybe add more understanding of why AMDKD is better than # models. Why training on mixed data is not as good as training separately then distilling? Any deeper understanding in the context of VRP?
3.  Why do the three distributions (Figure 1c-1e) generalize better to other distributions? Given a real-world application scenario where generalization needs to be improved, is there a strategy for including more distributions with AMDKD in addition to the existing distribution?

**Limitations:**

Yes

**Strengths And Weaknesses:**

 Strengths:
1. This paper presents a novel exploration of KD or multi-teacher KD for VRP;
2. The proposed AMDKD approach is simple but effective and could be easily applied to help improve the generalizability of existing models by training and distilling on multiple distributions;
3. The paper is very well written and easy to read.

Weaknesses:
1. There is some confusion about the time analysis.
2. Could provide more understanding of AMDKD.

---

> ### Author Response · Authors · 2022-08-02
> **Response to Reviewer #kcQQ (continued)**
>
>  **[Regarding the selection of exemplar distributions]**
>
> \- “*Why do the three distributions (Figure 1c-1e) generalize better to other distributions?* ” - We would like to clarify that our AMDKD is a generic framework that could train a given deep model (e.g., AM or POMO) on any numbers and types of exemplar distributions, and then generalize to other unseen distributions. We choose these three exemplar distributions (i.e., Uniform, Cluster and Mixed) because they are commonly used in the field of Operations Research to represent or resemble real-world datasets (e.g., in the benchmark datasets TSPLIB and CVRPLIB). Nevertheless, we have also presented the experimental results in Appendix C.1 (lines 602-611) to show that the above three exemplar distributions could be substituted with others, where our AMDKD framework still could consistently improve the cross-distribution generalization performance.
>
> \- “*Including more distributions*”- One could just simply sample instances from the real-world distribution (e.g., sampling based on the past operational data) and take them as a new additional exemplar distribution for training the AMDKD, to help improve the generalization in the real-world application scenario. On the other hand, since our AMDKD exhibits a favorable generalization performance on both unseen in-distribution and out-of-distribution instances, we believe that it has the potential to achieve good generalization in unseen datasets even without adding new exemplar distributions for re-training. In other words, we could directly apply the pre-trained AMDKD models in Section V to infer real-world instances.
>
> ---
> **Reference**
> ```
> [21] André Hottung, Yeong-Dae Kwon, and Kevin Tierney. Efficient active search for combinatorial optimization problems. In International Conference on Learning Representations, 2022.
> ```

---

> ### Author Response · Authors · 2022-08-02
> **Response to Reviewer #kcQQ**
>
> We appreciate the reviewer for the positive and insightful feedback. We are gratified that the reviewer found our AMDKD novel, effective, and could be easily applied to existing methods. The concerns from the reviewer are responded as follows.
>
> ---
> **[Regarding the time analysis]**
>
> By “*more efficient student model*”, it means that compared with the original teacher model (i.e. AM and POMO), our learned student model (i.e., AMDKD-AM and AMDKD-POMO) could not only reduce the model parameters but also consume less computation cost for inference. As shown in Table 2, the learned student model by AMDKD reduces the size of the teacher model from 0.68Mb to 0.26 Mb (a 61.8% reduction) for AM and from 1.20Mb to 0.49 Mb (a 59.2% reduction) for POMO; and our AMDKD-AM (13min) and AMDKD-POMO (1.1min) require less inference time compared with all the baselines including AM# (18min) and POMO# (1.3min).
>
> Regarding the “*EAS*” (efficient active search) in Table 2, it is a generic active search method that performs instance-dependent parameter updates during inference which was proposed in [21]. EAS can be applied to any trained deep models of construction type to achieve better performance with (much) longer inference time. Here, we mainly showcase that our AMDKD-POMO can also be coupled with EAS (a.k.a. AMDKD+EAS) during inference to further boost the performance when the users have a longer inference time budget, which is optional.
>
> ---
> **[Regarding the deeper understanding of AMDKD vs # model]**
>
> We present more possible intuitions/rationales/insights on why AMDKD works better than the # models as follows, which will also be discussed in the revised paper.
>
> - Different exemplar distributions may have different levels of difficulty for solving. In our AMDKD, the adaptive strategy allows the learning to dynamically concentrate more on the harder distributions that the student model may not tackle well, which is more efficient than simply mixing instances from different distributions into one training dataset as did in the training of # models. And this adaptive strategy echoes how humans learn knowledge, where more time is always required for harder subjects.
>
> - Without additional intervention, reinforcement learning tends to learn those easier things to get a higher reward. This means that training on mixed data (the # models) may possibly bias the learning towards distributions that are easier to solve or to get higher rewards. On the contrary, our AMDKD selects a specific teacher model based on the weakness of the current student model, which encourages it to learn those hard-to-solve distributions.
>
> - Directly training on a mixed dataset may not be efficient or stable. On the one hand, it would be hard for DRL to directly learn good patterns from mixed data that follow different distributions, due to the possibly limited representation capability of the neural networks for handling such hard optimization problems, even without the diversity in distributions. On the other hand, the tasks for different distributions may have different reward ranges (such as the route length), which may cause instability in the RL training. For example, the average total rewards for solving CVRP-100 instances following Uniform and Cluster distributions are around 16 and 7, respectively. Compared with # models, our AMDKD tackles this issue in a way that only one distribution is leveraged in a training epoch. Further empowered by the knowledge distillation, our AMDKD framework efficiently and effectively transfers useful knowledge (patterns) from various teacher models to a unified and light student model.
>
> - Finally, we note that it still remains an open issue and we will also explicitly investigate more about it in future.

---

### Official Review · Reviewer_6JzK · 2022-07-13

**Rating:** 5
**Confidence:** 5
**Soundness:** 3 good
**Presentation:** 3 good
**Contribution:** 3 good

**Summary:**

This paper proposes a knowledge distillation framework named Adaptive Multi-Distribution Knowledge Distillation (AMDKD) for transferring various knowledge from multiple expert teachers to a generalist lightweight student model.  Firstly, AMDKD performs teacher pre-training to obtain expert teachers for different distributions.  Then in each epoch of the knowledge distillation process, AMDKD selects a specific distribution and its teacher with an equal probability. The probability of each distribution being selected is computed according to the performance of the student model on this distribution.  With the selected teacher model, AMDKD performs a general single-teacher knowledge distillation and updates the student model. Several experiments on TSPLIB and CVRPLIB datasets are conducted and AMDKD shows competitive results compared with state-of-the-art methods.

**Questions:**

See the weaknesses.

**Limitations:**

See the weaknesses.

**Strengths And Weaknesses:**

Strengths：
1.	Benefitting from a concise description of the overall framework, the motivation and training process is clear and easy to read.
2.	The idea of selecting a specific teacher model on the weakness of the student model is interesting, which converts a multi-teacher knowledge distillation to a single-teacher knowledge distillation in each epoch, avoiding the profitless influence of averaging the outputs from multiple teacher models.
3.	Sufficient experiments are conducted to prove the effectiveness and the generalization of AMDKD, especially the experiments on the unseen in-distribution and out-of-distribution instances.

Weaknesses:
1.	The probability of selecting specific distribution and its teacher is updated with each epoch, which will take a lot of computation cost to evaluate the performance of the student model.
2.	The selection of teachers is decided by the performances on the validation datasets, which makes the experimental results rely on the quality of the validation datasets

---

> ### Author Response · Authors · 2022-08-02
> **Response to Reviewer #6JzK**
>
> We appreciate the reviewer for the positive and insightful feedback. We are also delighted that the reviewer found our proposed framework clear and interesting. The major concern about our adaptive strategy of teacher selection is responded as follows.
> ***
> **[Regarding the computation cost to evaluate the performance of the student model]**
>
> As pointed out by the reviewer in the strengths, our adaptive strategy prevents the profitless inference of averaging the outputs from different teachers. Through ablation studies in Table 4, we demonstrated that such design plays an important role to ensure the eventual good performance, where we also showed that our framework still works even without this adaptive rule (but of course the performance will become slightly inferior). The reviewer is correct that performing student model evaluation in each epoch will introduce additional computation cost. However, as ‘*there is no free lunch*’, we would like to argue that this is well acceptable for the following reasons, which we will also discuss in the revised manuscript.
>
> - **Such performance evaluation is only involved in the training but not the testing, and our proposed distillation framework actually leads to faster inference speed.** As shown in Table 2, AMDKD-AM (13min) and AMDKD-POMO (1.1min) require less inference time compared with the baseline model AM# (18min) and POMO# (1.3min). Given that the learned student model is lighter and faster with superior generalization capabilities, we believe such a trade-off between longer training time and higher inference performance is worthwhile.
>
> - **The increment in training time is acceptable.** Taking AMDKD-AM training on CVRP-100 as an example, the total evaluation time is 2.7s (0.9s per exemplar distribution on average) for each epoch, which is approximately 1% of the total training time (i.e., 4 min, line 239). We apologize for the typo (0.4 min) in the original manuscript, which will be rectified.
>
> - **We may stop the student evaluation early for even faster training if needed.** Inspired by the comment of the reviewer, we quickly conduct additional experiments and find that the probability of selecting different teacher models would eventually converge to a stable one. This means that we may stop the evaluation early to speed up the training further if needed. For example, for training AMDKD-AM on CVRP-100 (see Figure 6 in the revised appendix), we may stop the student evaluation early at around 3,000 epochs.
>
> ***
> **[Regarding the reliance on the quality of the validation datasets]**
>
> We would like to clarify that the used validation datasets are all randomly generated using exactly the same method for generating the training datasets, which indicates that the validation datasets are easy to collect and the dataset quality is basically consistent with the training ones. In this sense, we believe the number of instances in the validation dataset may somehow capture the quality of the dataset. As mentioned in the above response, there is a trade-off between the training time and the validation time. Therefore, we set the size of the validation dataset as 1,000 (such setting is also used in [10] to evaluate the performance of the proposed model). We will add additional experiments to further verify and provide more insights on the impacts of the validation datasets (which may need a couple of weeks to implement and run the experiments). However, based on our experience, such influence might be limited given that, sampling from the same training distributions, and the quality of the validation datasets may influence how fast the probabilities converge (if we refer again to Figure 6 in the revised appendix), but not the final performance. Nevertheless, we agree that it is interesting and meaningful to explore more advanced designs on how to create even better validation datasets. We leave this as an important future work since 1) our current main focus is to propose a generic framework for learning generalizable routing models, and 2) our randomly generated validation datasets look sufficiently effective in achieving favorable performance eventually.
>
> ---
> **Reference**
> ```
> [10] Mohammadreza Nazari, Afshin Oroojlooy, Martin Takác, and Lawrence V Snyder. Reinforcement learning for solving the vehicle routing problem. In Advances in Neural Information Processing Systems, pages 9861–9871, 2018.
> ```

---

> > ### Comment · Reviewer_6JzK · 2022-08-09
> > **Thanks for the detailed response**
> >
> > Thanks for the detailed response which solved my concerns about the computation cost and the reliance on the validation dataset. However, after reading this response, I still have the following comments:
> >
> > A. As shown in Table 2, AMDKD is indeed faster than the baseline model in inference. But I can’t ignore the difference in model sizes. The size of the model used in AMDKD-AM (0.26Mb) is smaller compared with the baseline model AM# (0.68Mb), which is helpful for the reduction of inference time. So I think these comparisons are kind of unfair. Whether the baseline methods can maintain similar performances on the small models remains to be proved by experiments.
> >
> > B. Randomly sampling the validation datasets from the same distribution of the training datasets is indeed helpful for ensuring the quality of the validation datasets. But to control the training time, the sizes of validation datasets cannot be too large. So when the training datasets get bigger and bigger and their distributions become more and more complex, I think it’s hard for the limited validation datasets to describe the characteristics of the training datasets' distributions, which will affect the performance of this method, especially when the distributions of training datasets part overlap each other. Some dataset distillation methods may be useful for generating the validation datasets. However, how to deal with the balance between the training time and the sizes of validation datasets is still a problem.
> >
> > Due to the tight schedule, it seems to be impossible to answer all these questions correctly during the discussion period. So the initial rate is preserved.

---

> > > ### Author Response · Authors · 2022-08-09
> > > **Thanks for the comments and please kindly check our further clarifications (continued)**
> > >
> > > **[Regarding Comment B]**
> > >
> > > We thank the reviewer for acknowledging that randomly sampling the validation datasets from the same distribution of training is helpful for ensuring the quality of the validation datasets. And we agree that the validation dataset cannot be too large (In fact, ours is not too large given that the induced extra computation time for validation is limited, where as acknowledged by the reviewer, the concern about the computation cost has been addressed).
> > >
> > > **However, we would like to argue that the size of our training datasets is NOT “*getting bigger and bigger and more and more complex*” for the following reasons.**
> > >
> > > - We are using on-policy RL methods (REINFORCE) rather than off-policy RL methods (e.g., DQN), where we do not have a replay buffer to store past training data. In fact, the size of our training data is fixed, and they are randomly generated on the fly within each epoch. This means that past training data will be discarded immediately after use (since REINFORCE requires all training data to be generated online), and new training data (of the same amount) are randomly generated for any new epoch. We also clarify that the quality/complexity/hardness of the training data in the first epoch is essentially the same to that in each subsequent epoch, since they are all sampled from the same pre-defined distributions (e.g., Uniform, Cluster, or Mixed). Therefore, the training datasets are not growing larger and larger or getting more and more complex.
> > >
> > > - Regarding “*especially when the distributions of training datasets part overlap each other*”, we would like to clarify that this may not happen in our framework. Compared with training on instances following mixed distributions (i.e. the overlap of distribution mentioned by the reviewer), our AMDKD leverages only one training distribution within each epoch and performs single-teacher KD in turns instead of multi-teacher KD at the same time, which avoids the profitless influence of averaging the outputs of multiple teachers (as acknowledged by the reviewer). We also kindly refer the reviewer to our response *[Regarding the deeper understanding of AMDKD vs # model]* to reviewer #kcQQ.
> > >
> > > - Meanwhile, by referring to the term “*training data*” in RL, it is actually the sample records of the agent interacting with the environment rather than a fixed or a growing training dataset that is directly used to train the model like in supervised learning. In this regard, the above mentioned training/validation datasets are just a pool of tasks (i.e., VRP instances) used to generate real training samples (i.e., the trajectories with rewards, which are more important to reflect the performance of the model). To correctly evaluate the quality of the sampled trajectories, we used a strong solver LKH (see line 201) to calculate the (optimality) gaps defined in Eq. (5), which we believe also helps make the validation performance smooth and effective.
> > >
> > > Therefore, we believe that even the somehow “*limited*” validation dataset could still basically capture the performance of the student model (as demonstrated by the good performance of AMDKD). Nevertheless, we fully agree with the reviewer that dataset distillation techniques that study the representativeness/hardness of the instances (even within the same distributions) would be another interesting topic. Since we focus more on training a single model to handle instances from different distributions, we will consider this valuable advice as important future work, for which we will also discuss in the revised paper.
> > >
> > > Finally, regarding “*the balance between the training time and the sizes of validation datasets is still a problem*”, we do agree that there should be a tradeoff. As also acknowledged by the reviewer, ‘*there is no free lunch*’,  and we will include more results in the revised paper to discuss the pros and cons. We thank the reviewer again for those valuable concerns and comments.

---

> > > ### Author Response · Authors · 2022-08-09
> > > **Thanks for the comments and please kindly check our further clarifications**
> > >
> > > We thank the reviewer for the detailed reply. We are glad that our response addressed most of the concerns about the computation cost and the reliance on the validation datasets. Regarding the further concerns/comments, please see our response below where we believe there might be some misunderstandings.
> > >
> > > ---
> > >
> > > **[Regarding Comment A]**
> > >
> > > As mentioned by the reviewer, our AMDKD is faster since our AMDKD-AM student has a smaller network (dim = 64, 0.26Mb) than the baseline AM# (dim = 128, 0.68Mb). We understand the concern about the ‘unfair’ comparison; however, **we would like to argue that this is the advantages of our method rather than limitations.**
> > >
> > > - On the one hand, the presented results in Table 6 (in Appendix) and Appendix C.1 have somehow already clarified this concern. We showed that when also reducing the network dimension of AM# from 128 to 64 (as our student model), AM# (64) performs significantly inferior to ours (the same case for POMO# and AMDKD-POMO). Thus, it suggests that it is hard to directly learn a powerful smaller model via reinforcement learning (RL) in handling various distributions, and our AMDKD (via knowledge distillation) is more effective and performs more favorably against the baselines.
> > >
> > > - On the other hand, as shown in Figure 3b, if we do not reduce the model size of our student model, i.e., let the student (64) share the same architecture (model size) with its teachers (128), our AMDKD could be further boosted with even better (generalization) performance.
> > >
> > > - Therefore, we emphasize that it is the advantage of our distillation framework AMDKD that allows us to reduce the dimensionality of the network while preserving the favorable performance of the (large) original model on its default distribution (with faster inference speed) and, more importantly, boosting the capability of the (small) student model in generalizing across different distributions. However, we appreciate the concern of the reviewer, and will explicitly mention that the models have different sizes of parameters when analyzing the runtime in Table 2.

---

### Official Review · Reviewer_RL4k · 2022-08-27

**Rating:** 7
**Confidence:** 4
**Soundness:** 4 excellent
**Presentation:** 3 good
**Contribution:** 3 good

**Summary:**

The paper focuses on a class of NP-hard combinatorial optimization problems called Vehicle Routing Problems (VRP) which is of wide practical interest. In the deep learning community there is a wide interest in replacing the domain and expert dependent heuristics with data-driven methods like reinforcement learning. However, as the paper notes, most current approaches focus on one particular distribution of problem rather than trying to generalize across distributions, which limits their practical applicability. The paper builds on the ideas of knowledge distillation widely applied in practical deep learning applications to train a light-weight student model to learn from this variety of individual distributions focused methods. This allows demonstration of competitive results on both unseen in-distribution as well as OOD instances while requiring cheaper compute resources.

**Questions:**

Would be useful to understand the computational tradeoff for the larger student models as well. What's their resulting size and time taken?

**Limitations:**

The paper adequately mentions its limitations.

**Strengths And Weaknesses:**

### Strengths

A simple and widely used idea of knowledge distillation is demonstrated useful in a new domain with wide practical applications. Given the simplicity the method is likely simple to implement and generalize to real world problems. The shown improvements are fairly consistent along a variety of different axes of comparisons and are compared against a fair number of strong baselines. They also confirm that their method can improve further with more compute resources for their student model.

### Weaknesses

Main weakness as the authors acknowledge as future work is generalizing across different/larger problem sizes. The idea of knowledge distillation is straightforward and not necessarily novel, but this will likely be a strong baseline for the community to build upon.

---

### Author Response · Authors · 2022-08-08
**General Response**

We thank the reviewers for the acknowledgement and support to our work. Since the author-reviewer discussion deadline is approaching, and it would be highly appreciated if the reviewers kindly spare more time for us to respond to the further comments if they still have. Thanks a lot.

---

### Meta-Review · Area_Chair_JnWT · 2022-08-27

**Recommendation:** Accept
**Confidence:** Certain

**Metareview:**

All the reviewers are in agreement to accept the paper. The paper tackles vehicle route planning via a knowledge distillation framework using student teacher models. The ideas in the paper are appreciated by all the reviewers. There are minor criticisms, especially regarding scalability with the problem size, selection of teachers, and the fact that the knowledge distillation is relatively known.

Given the minor criticisms from the original three reviewers, I asked for another expert in the field to look at it and the reviewer agreed with the rest of the pool. Additionally, I took a good look at the paper as well and I am happy to recommend accepting the manuscript.


**Award:**

No

---

### Decision · Program_Chairs · 2022-09-14

Accept